# Convolutional Visual Prompt
# for Robust Visual Perception

**Yun-Yun Tsai**[*]
Columbia University
yunyuntsai@cs.columbia.edu

**Chengzhi Mao**[*]
Columbia University
mcz@cs.columbia.edu

**Junfeng Yang**
Columbia University
junfeng@cs.columbia.edu

## Abstract

Vision models are often vulnerable to out-of-distribution (OOD) samples, which often need adaptation to fix them. While visual prompts offer a lightweight method of input-space adaptation for large-scale vision models, they rely on a high-dimensional additive vector and labeled data. This leads to overfitting when adapting models in a self-supervised test-time setting without labels. We introduce convolutional visual prompts (*CVP*) for label-free test-time adaptation for robust visual perception. The structured nature of CVP demands fewer trainable parameters, less than 1% compared to standard visual prompts, combating overfitting. Extensive experiments and analysis on a wide variety of OOD visual perception tasks show that our approach is effective, improving robustness by up to 5.87% over several large-scale models.

## 1 Introduction

Deep models surpass humans when tested on in-distribution data, yet their performance plummets when encountering unforeseen out-of-distribution (OOD) data at test time, such as unexpected corruptions and shiftings [20, 22, 21, 18, 44]. This vulnerability raises serious risks when these models are deployed, especially in safety-critical applications and high-stakes tasks [49]. Prior works have studied how to improve generalization to OOD data at training time [54, 38, 56, 12, 36, 35, 39, 31], yet little work has been done to adapt the model to OOD at test time, and most of them require to modify model weights [59, 71, 55, 33].

Visual prompting emerges as an efficient and lightweight method to adapt the model at test time without modifying the model [11, 60, 1] (previously also called *adversarial reprogramming*). In contrast to finetuning the whole model weights, prompting can modify the original task of the model by providing context in the input space. It requires fewer OOD samples and simplifies model version management for practical applications. However, the OOD samples must be labeled, preventing these methods from combating unforeseen distribution shifts.

Recent work [40] defends against unseen attacks at test time by repairing the adversarial inputs with "reversal vectors" – high-dimensional prompt vectors directly added to inputs. It generates the prompts by minimizing a self-supervised loss, requiring no prior labeled OOD samples. Unlike adversarial attacks that damage arbitrary pixels by arbitrary amounts (within a given bound), however, structured changes known as visual distribution shifts are not effectively addressed by unstructured high dimensional vector prompts. Given that a self-supervised objective often has a shortcut and trivial solutions [13], we empirically find prompts without the right structures often improve performance minimally. (Sec. 5.2).

This paper presents **Convolutional Visual Prompt (CVP)**, which uses the convolutional structure as an inductive bias for adapting to visual OOD samples at test time. Prior work has demonstrated

---

[*]equal contributions

37th Conference on Neural Information Processing Systems (NeurIPS 2023).

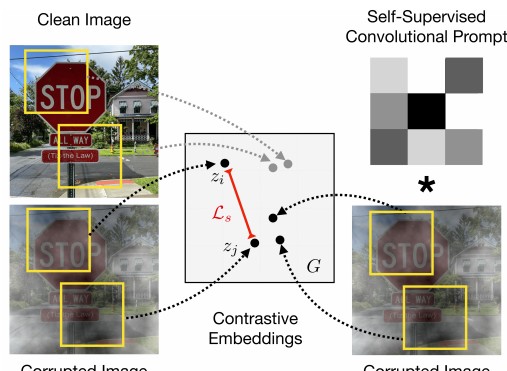

Clean Image

Self-Supervised Convolutional Prompt

Corrupted Image

Contrastive Embeddings

Corrupted Image

Figure 1: We demonstrate the self-supervised convolutional prompt that adapts model for robust perception. Clean image has a low self-supervision loss, denoted by the distance $\mathcal{L}_s$ on the contrastive embedding. When an image is corrupted (e.g, a stop sign in foggy weather), the self-supervised loss generally increases. We then apply the convolutional visual prompts on the corrupted samples, which instruct the model to adapt and produce a small self-supervision loss.

that convolutional operations are effective for handling structured data with local motifs [30, 64]. Inspired by this, CVPs are convolutional kernels with only a small number of tunable parameters – less than 1% of the number of tunable parameters in typical unstructured visual prompts (see Figure 1 for illustration), making CVP extremely lightweight and efficient to adapt.

Our experiments show the importance of structured inductive bias for self-supervised adaptation: compared to high-dimensional free-form visual prompts, standard low-rank structured prompts improve robustness by 3.38%, and convolutional structure in prompts is significantly better than low-rank prompts by 2.94% on CIFAR-10-C. On ImageNet-Rendition, ImageNet-Sketch, and 15 types of unforeseen corruptions for both CIFAR-10 and ImageNet at five different severity levels, CVP improves robustness by 5.87% for popular large-scale visual models, including ResNet50, WideResNet18, and the state-of-the-art vision-language model CLIP [51]. Since our method modifies the input space, it complements established test-time model weight adaptation methods (e.g., TENT [62], BN [55], and MEMO [71]) and can be generalized to multiple self-supervised objectives, such as contrastive learning [4], rotation [14], and masked autoencoder (MAE) [15].

## 2 Related Work

**Domain Generalization.** OOD data can lead to a severe drop in performance for machine learning models [20, 18, 53, 44, 39, 42]. Domain generalization (DG) aims at adapting the model with OOD samples without knowing the target domain data during training time. Adapting the model on OOD data [74, 10, 32, 73, 71, 55, 39, 42, 62, 59] also improves robustness.

Test-time adaptation is a new paradigm for robustness against distribution shift [40, 59, 71], mostly updating the weights of deep models. BN [55, 33] updates the model using batch normalization statistics, TENT [59] adapts the model weight by minimizing the conditional entropy on every batch. TTT [59] attempts to train the model with an auxiliary self-supervision model for rotation prediction and utilize the SSL loss to adapt the model. MEMO [71] augments a single sample and adapts the model with the marginal entropy of the augmented samples. Test time transformation ensembling (TTE) [50] proposes to augment the image with a fixed set of transformations and ensembles the outputs through averaging. The only two works that do not update the models is [40, 45], which modifies the pixels of adversarial, not OOD, samples to minimize the self-supervised objective.

**Visual Prompting.** Prompting was proposed in the natural language processing field to provide context to adapt the model for specific tasks [2]. Leveraging this idea, visual prompts [68, 26] adapt the model with a small number of trainable parameters in input space for vision tasks [9, 43] and foundation models [51]. Others proposed to prompt the samples with adversarial perturbations to repurpose the model for target classification tasks, known as *adversarial reprogramming* [11, 27, 60, 66], sharing the same idea with Visual Prompting. Black-box adversarial reprogramming [60] reprograms a black-box model for downstream classification tasks with limited data. V2S [66] reprograms speech recognition model for time-series data classification tasks. Robust visual prompts [41] are tuned at training time to improve the adversarial robustness of the model under attack. However, this work has not yet been applied to domain generalization where distribution is naturally shifted.

**Self-supervised learning (SSL).** SSL can learn effective representations from images without annotations [7, 6, 3, 22]. Prior works have shown that representations learned from different pretext tasks

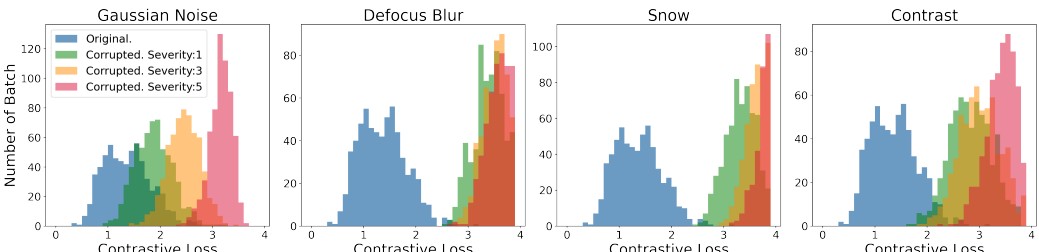

Figure 2: We show the histogram of the contrastive loss distribution on different corruption types. The blue region represents the loss distribution of original sample. The yellow, green, and red regions represent the loss distribution of corrupted samples with different severity (1, 3, and 5). Our plot shows the great shifting in SSL loss distribution between original and corrupted samples.

(jigsaw puzzles [47], rotation prediction [14], image colorization [29] and deep clustering [25], etc.) can be leveraged for several downstream tasks such as image classification [4], object detection [8] and test-time domain adaptation [58]. Another well-known branch of SSL is contrastive learning, which aims at grouping associated features for a transformation of samples and distancing from other samples with dissimilar features in the dataset [5, 16, 48]. Some of the methods [22, 57, 70] use SSL for outlier detection, which aims to learn generalizable out-of-distribution features and rejects them during the testing time. In contrast to these methods which require information from the targeted domain distribution during the training phase for adaptation, our method can adapt the model without requiring any information on unforeseen domains during the testing phase.

## 3 Test-Time Convolutional Visual Prompting

### 3.1 Learning Intrinsic Structure with Self-Supervision Task

Standard ways to improve model robustness on OOD data is through making the training robust, where the training algorithm anticipates possible corruptions and distribution shifts at inference time and trains on them [18]. However, anticipating the test-time shifting is a strong assumption which is often unrealistic in the real world. Therefore, we improve robustness at test time by dynamically adapting to unforeseen corruptions and unknown shifts.

The ideal case of adapting the model at inference time is to know the ground truth label for the target task, yet this is impossible given that the test data is not labeled. To significantly improve performance on the downstream classification tasks for unlabeled data, we must choose the right self-supervision task that shares rich information with the target classification task.

There is a large body of work on good self-supervision tasks for representation learning at training time. For example, jigsaw puzzles [47], rotation prediction [14], image colorization [29] and deep clustering [25] can be applied to several downstream tasks such as image classification [4], object detection [8] and test-time domain adaptation [58].

For visual recognition, a popular self-supervised objective is contrastive learning, which learns a representation that maps the feature of the transformations of the same image into a nearby place. This is formally defined as:

$$\mathcal{L}_s(x) = -\mathbb{E}_{i,j}\left[ y_{i,j}^s \log \frac{\exp(\cos(z_i, z_j))/\tau}{\sum_k \exp(\cos(z_i, z_k))/\tau} \right], \tag{1}$$

where $z$ are the contrastive features of $x$ extracted from pre-trained backbone. $y_{i,j}^s$ is a 0-1 vector for indicating the positive pairs and negative pairs. If $y_{i,j}^s$ is 1, the $i$-th feature $z_i$ and $j$-th feature $z_j$ are both from the $x$ sample. Otherwise, they are from different $x$. We denote $\cos(\cdot, \cdot)$ the cosine similarity, $\tau$ as temperature scaling value. We optimize the model parameters for SSL model $\mathcal{C}$ by using the contrastive loss. The objective function for training is defined as $\min_{\theta_{\mathcal{C}}} \mathbb{E}_{(x)\sim\mathcal{X}_s}\left[\mathcal{L}_s(\cdot)\right]$, where $\mathcal{X}_s$ is the source domain data for training. We train only on the clean samples drawn from the non-corrupted dataset. We compare this task to other self-supervision tasks in our ablation study.

Since we need to train the self-supervised task at training time, the self-supervised task will perform high on the test data that is the same distribution as the training one. Our results find that the

performance of the self-supervised task drops largely when the distribution is shifted at test time. See Figure 2. This suggests that the information that is useful for the self-supervised task is collaterally corrupted in addition to the classification performance drop.

Prior work demonstrated significant mutual information between self-supervised tasks and perception tasks, where pretraining with the self-supervised task improves the perception tasks' performance [7, 6, 3, 22]. We propose to adapt the model to minimize the self-supervised loss at inference time, where self-supervised task is a proxy that captures a large portion of information from the perception task. In recovering the information of the self-supervised tasks, we could recover the information of perception tasks that was corrupted due to distribution shifts.

## 3.2 Test-time Adaptation for Vision Models

Inference time adaptation allows the model to adapt to the unique characteristics of the new distribution online. However, the key challenge is that the SSL objective we use to adapt the model is a proxy, where there is often a trivial solution that reduces the loss but adapts the model in the wrong way. Several established methods exist to adapt the vision models, including foundation models.

**Finetuning (FT)** is a standard way to adapt the deep model. It often optimizes all the parameters of the deep models or partially, which is often heavy and requires large space to save the copy of the model to re-initialize and update. Prior work shows that this method is effective when it is finetuned on supervised tasks, yet adapting with the self-supervised task on a few examples remains under-explored. We discuss the finetuning method using the self-supervised task.

**Partial Finetuning (PFT)** is another way to adapt the model at inference time by only changing the statistics of the batch-normalization layer. This method assumes that the distribution drifts in the mean and standard deviation of the test data and can be recovered through test-time adaptation. The closest existing works are BN [55], Tent [62] and MEMO [71]. Tent updates the BN statistics but needs to continue training on the same distribution. MEMO only requires a single test data point, yet the algorithm is slow due to the whole model update and heavy augmentations. Here, we adapt batch normalization through our proposed contrastive learning-based self-supervised loss.

**Visual Prompts (VP)** have emerged as a lightweight way to adapt pre-trained models. There are two major ways to apply visual prompts to the vision model. Let the image be $\mathbf{x}$, it adds a vector $\mathbf{v}$ to the input image: $\mathbf{x} = \mathbf{x} + \mathbf{v}$. For low-rank visual prompts [24, 67], we use a low-rank matrix in $v$ during optimization. Most visual prompts are studied in training setup, yet they are under-explored at inference time. [40, 45] optimize an additive visual prompt on the image to repair the adversarial perturbation and improve the model robustness.

## 3.3 Adapting via Convolutional Visual Prompts

Adding the right structure is an effective way to avoid trivial solution to the self-supervised objective. We now introduce the convolution visual prompts (CVP), which instruct the deep model to adapt to the test distribution through convolution. Convolution has been proved to be a successful inductive bias for visual tasks [28], which is more sample efficient [34]. Our assumption is that a large family of distribution shifts in the image data are visually structured, which can be modeled by convolution. Our prompt is simple and is defined as:

$$\mathbf{x} = \mathbf{x} + \lambda \text{conv}(\mathbf{x}, \mathbf{k}) \tag{2}$$

One major advantage of the convolution prompt is that the number of parameters in the prompt is significantly lower (1%) than other conventional prompts (e.g., patch prompt and padding prompt). Compared to adapting the whole model weights, our method is light weight and is fast by exploiting the structure of visual distribution shifts In Appendix 7.2, we show the detailed algorithm of CVP. Our lightweight adjustment allows the model to quickly update to the novel data points without too much computation and memory overhead. As vision models are continuously deployed in edge devices, this is extremely important, given the limited computational resources.

# 4 Experiment

This section demonstrates the detail of experiment settings and evaluates the performance of our method CVP, compared with conventional Visual Prompts (VP) and existing test-time approaches.

More analysis are shown in Section 5 and Appendix. We do a comprehensive study on the CVP, including different prompts design, kernel v.s. structure analysis, multiple SSL tasks, sensitivity analysis on the batch size and adapt iterations, GradCam visualization, and optimization cost of CVP.

## 4.1 Experiment Setting

**Dataset.** We evaluate our method on five kinds of OOD datasets, including CIFAR-10-C [21], ImageNet-C [46], ImageNet-R [19], ImageNet-Sketch [65], and ImageNet-A [23]. The following describes the details of all datasets.

• **Synthetic OOD Data.** The corruption data are synthesized with different types of transformations (e.g., snow, brightness, contrast) to simulate real-world corruption. The dataset contains CIFAR-10-C and ImageNet-C. Both of them are the corrupted versions of their original dataset, including 15 corruption types and 5 severity levels. A larger severity level means more corruption is added to the data. To well evaluate our method, We generate the corruption samples with five severities based on the official GitHub code[*] for each 15 corruption types.

• **Natural OOD Data.** The ImageNet-Rendition [18] contains 30000 images collected from Flickr with specific types of ImageNet's 200 object classes. ImageNet-Sketch [65] data set consists of 50000 sketch images. The ImageNet-Adversarial [23] is a natural adversarial shifting dataset contains 7500 images collected from the natural world.

**Model.** The backbone model architecture is pretrained on WideResNet18 [69] and ResNet26 [17] for CIFAR-10-C, ResNet50 [17] for ImageNet-C, Rendition, and Sketch. We extract the logit features before the fully connected layer of the backbone model for training the SSL model. The SSL model is a simple MLP with the final layer outputting the one-dimensional features for the contrastive learning task. We further extend our prompt method to the foundation model CLIP [51], where we only prompt the vision encoder.

**Baseline Details** We compare several test-time adaptation benchmarks with CVP.

• **Standard**: The baseline uses the pre-trained model without adaptation. For CIFAR-10-C, the standard is trained with 50000 clean CIFAR-10 train dataset on WideResNet18 and ResNet. For ImageNet1K-C, the standard is trained with $\sim$1.2M clean ImageNet train dataset on ResNet50.

• **Finetune (FT)**: We adjust the whole model weight for every upcoming batch during the inference time with the self-supervised loss. In our experiments, after one-batch fine-tuning, the model will be restored to the initial weight status and receive a new type of corrupted samples.

• **Partial Finetune (PFT)**: The partial fine-tune adapts batches of the samples to the model by only adjusting the batch normalization layers with self-supervised loss at every inference time. Same as Finetune baseline, the model will be restored to the initial weight status after the one-batch adaptation.

• **SVP [40]**: The prompting method to reverse the adversarial attacks by modifying adversarial samples with $\ell_p$-norm perturbations, where the perturbations are also optimized via contrastive loss. We extend this method with two different prompt settings: patch and padding. For the patch setup, we directly add a full-size patch of perturbation into the input. For the padding setup, we embed a frame of the perturbation outside the input. More baseline detailed are shown in the Appendix 7.3

**Design of Convolutional Visual Prompts (CVP)** We prompt the input samples by adding the convolutional kernels. Our kernels can be optimized under several different settings, including 1.) fixed or random kernel initialization 2.) 3*3 or 5*5 kernel sizes. We show a detailed evaluation of all kernel setups in the experimental results. For initialization, we can either random initialize the kernel size $k$ in a uniform distribution or initialize with fixed values. For fixed initialization, we empirically found that starting from a sharpness kernel is effective. We optimize the kernel with 1 to 5 iterations of projected gradient descent. To preserve the original structure, we combine the residual of input and convolved output with learnable parameters $\lambda$. We jointly optimize the convolutional kernel $k$ and $\lambda$ with the self-supervised loss $\mathcal{L}_s$. The range of $\lambda$ is predefined and empirically set up in a fixed

---

[*]We generate all types of corruption data based on the GitHub code: `https://github.com/bethgelab/imagecorruptions`

| Method/Model | WideResNet18 Avg. Error (%) | ResNet26 Avg. Error (%) |
|---|---|---|
| Standard | 58.24 | 62.12 |
| FT | 69.33 (+11.09) | 63.07 (+0.95) |
| PFT | 62.65. (+4.41) | 62.47 (+0.34) |
| SVP (patch) | 57.94 (-0.3) | 62.82 (+0.7) |
| SVP (padding) | 58.20 (-0.04) | 62.18 (+0.06) |
| CVP-F3 | 53.67 (-4.57) | 59.52 (-2.6) |
| CVP-R3 | **52.37 (-5.87)** | **59.32 (-2.80)** |

Table 1: Corruption benchmarks on CIFAR-10-C. We show the average error rate on 15 corruption types and 5 severities for CVP, compared with the baselines. For the CVP results, we denote the two initialization settings: fixed/random as F / R, and the number behind F / R means the kernel size (e.g., 3 or 5). We set up the batch size of adaption as 16 and the number of adapt iters as 5 for all methods.

| | ImageNet-C mCE ↓ | ImageNet-R Error (%) | ImageNet-S Error (%) | ImageNet-A Error (%) |
|---|---|---|---|---|
| ResNet50 | 76.87 | 63.83 | 75.90 | 100.0 |
| FT | 77.10 (+0.26) | 64.38(+0.55) | 76.38 (+0.48) | 99.95 (-0.05) |
| PFT | 76.74 (-0.13) | 69.63 (+5.8) | 80.43 (+4.53) | 99.89 (-0.11) |
| SVP (patch) | 76.74 (-0.13) | 68.86 (+5.03) | 75.93 (+0.03) | 99.94 (-0.06) |
| SVP (padding) | 80.07 (+3.23) | 63.84 (+0.01) | 75.92 (+0.02) | 99.91 (-0.01) |
| CVP-F3 | 75.88 (-0.95) | 63.56 (-0.27) | 75.32 (-0.58) | 99.2 (-0.8) |
| CVP-R3 | **75.34 (-1.49)** | 63.49 (-0.34) | **75.30 (-0.60)** | 99.2 (-0.8) |
| CVP-F5 | 75.74 (-1.09) | 63.18 (-0.65) | 75.35 (-0.55) | 98.67 (-1.33) |
| CVP-R5 | 75.77(-1.06) | **63.06 (-0.77)** | 75.33 (-0.57) | **98.4 (-1.6)** |

Table 2: Comparison on CVP with other baselines on ImageNet-C, ImageNet-R, ImageNet-S, and ImageNet-A. The standard model is ResNet50 [17]. For ImageNet-C, we calculate the mean corruption error (mCE). Same as Table 1, we evaluate the CVP under multiple kernel setups and show the results. For ImageNet-C and Sketch, CVP-F3 achieves the best robustness, which reduces the error rate by 1.49% and 0.6%. Other datasets, such as ImageNet-R and ImageNet-A, outperform all the baselines when updating 5*5 kernels with random initialization.

range. For CIFAR-10-C, we set up the range as [0.5, 3], and for ImageNet-C is set up as [0.5, 1]. We describe the detailed parameter settings and different prompt designs in the Appendix 7.4 7.7.

## 4.2 Experimental Results

Table 1 presents the evaluation results of CVP on CIFAR-10-C. We compare CVP with five different baselines, including standard, VP (patch/padding), fine-tune (FT), and partially fine-tune (PFT) with SSL loss, using two model architectures. We also explore different kernel settings of CVP, including fixed/random initialization and different kernel size. Results show that for WideResNet18, CVP achieves the highest reduction in average error rate by 5.87% when updating the 3*3 kernel with random initialization. For ResNet-26, CVP consistently reduces the error rate by 2.8%.

Table 2 displays the results for ImageNet-C, Rendition, Sketch, and Adversarial. The standard baseline is pre-trained on ResNet50 [17]. For ImageNet-C, we report the mean corruption error (mCE), calculated with the performance rates on ResNet50 and standard AlexNet following the benchmark [21]. Due to the larger dimension of the ImageNet dataset (224), a larger kernel size effectively captures more information from the input. Thus, we set the kernel size as 3*3 and 5*5. Our findings indicate that CVP achieves the highest error rate reduction for ImageNet-R and A by 0.77% and 1.6%, respectively, when updating with a 5*5 kernel using random initialization. CVP consistently outperforms all baselines for ImageNet-C and Sketch. We observe that VP (patch) and VP (padding) degrade performance on most datasets, suggesting that unstructured perturbations with more trainable parameters are ineffective in adapting to natural OOD data with general shifts.

In Table 3, we further evaluate the performance of CVP on the large-scale foundation model, CLIP [52]. Compared to the baselines VP (padding) and VP (patch), CVP achieves the best performance on all datasets when updating under random initialization with 5*5 kernel size.

**CVP Complements Other Test-Time Adaptation Methods** Since our method modifies the input space and aligns the representation of adapted samples to the original manifold, it also complements established test-time adaptation approaches which adjusts the model weights. Thus, we combine the CVP with several existing methods, including MEMO [71], BN [55], and TENT [62]. For a fair comparison, we set up the batch size as 16 for all experiments. for the parameter settings, we set the kernel size for CIFAR-10-C as 3x3 and ImageNet-C, R, S, and A as 5x5. The adaptation iterations

|  | CIFAR-10-C Avg. Error (%) | ImageNet-C mCE ↓ | ImageNet-R Error (%) | ImageNet-S Error (%) |
|---|---|---|---|---|
| **CLIP(ViT/32)** | 58.39 | 77.93 | 32.09 | 60.55 |
| SVP (patch) | 58.43 (+0.04) | 77.81 (-0.12) | 32.12 (+0.03) | 60.53 (-0.02) |
| CVP-F3 | 57.94 (-0.45) | 77.43 (-0.50) | 31.16 (-0.93) | 59.47 (-1.08) |
| CVP-R3 | 57.91 (-0.48) | 77.71 (-0.22) | 31.31 (-0.78) | 59.62 (-0.93) |
| CVP-F5 | 57.98(-0.41) | 77.25 (-0.68) | 31.14 (-0.95) | 59.83 (-0.72) |
| CVP-R5 | **57.79 (-0.60)** | **76.67 (-1.26)** | **30.43 (-1.66)** | **59.43 (-1.12)** |

Table 3: Evaluation on the CLIP model. We compare CVP with other prompting baselines on CIFAR-10-C, ImageNet-C, ImageNet-R, and ImageNet-S. Overall, the CVP-R5[†] achieves the best performance, which reduces the error rate by 1.16% on average.

|  | CIFAR-10-C Avg. Error (%) | ImageNet-C mCE ↓ | ImageNet-R Error (%) | ImageNet-S Error (%) | ImageNet-A Error (%) |
|---|---|---|---|---|---|
| Standard | 58.24 | 76.87 | 63.83 | 75.90 | 100.0 |
| Mao et al. [40] | 57.94 (-0.3) | 76.74 (-0.13) | 68.86 (+5.03) | 75.93 (+0.03) | 99.94 (-0.06) |
| CVP (ours) | 52.37 (-5.87) | 75.43 (-1.49) | 63.06 (-0.77) | 75.30 (-0.6) | 98.4 (-1.6) |
| MEMO [71] | 56.14 | 73.45 | 60.73 | 73.43 | 99.1 |
| MEMO + CVP | 54.84 (-1.3) | 72.02 (-1.43) | 60.23 (-0.5) | 72.67 (-0.76) | 98.64 (-0.46) |
| BN [55] | 38.51 | 76.20 | 67.29 | 77.98 | 99.8 |
| BN + CVP | 37.39 (-1.12) | 76.16 (-0.04) | 67.21 (-0.08) | 77.92 (-0.06) | 98.67 (-1.13) |
| TENT [62] | 38.52 | 70.45 | 58.45 | 73.88 | 99.7 |
| TENT + CVP | **36.69 (-1.83)** | **70.34 (-0.11)** | **58.42 (-0.03)** | **73.83 (-0.05)** | **98.54 (-1.16)** |

Table 4: Our prompt method complements other test-time adaptation approaches that update model weight, including MEMO, TENT, and BN. We show complements gain on every baseline when combined with CVP. Here, the Standard for CIFAR-10-C is WideResNet18 and other dataset is ResNet50. For CIFAR-10-C, on top of the TENT [62] method, we achieve the best gain on the performance, which reduces 1.83% error rate.

are all set as 5. In Table 4, we show the results on five benchmarks. For CIFAR-10-C, CVP improves 1.83 points on top of TENT and reduces the error rate by 21.55% compared with the standard one. For the other datasets, CVP achieves the lowest error rate on top of the TENT method. However, the BN method degrades the performance under the small batch setting. Due to the page limit, in Appendix 7.5, we show more evaluations on other benchmark, such as the Cutout-and-Paste data [61] and other baseline Test Time Training (TTT) [59].

# 5 Ablation Study

**Low-Rank Structure to Avoid Shortcut in SSL Objective.** Since SSL objective is a proxy for our visual recognition task, sole minimizing SSL objective can produce a overfitted prompt without much improvement in visual recognition. A typical way to avoid this suboptimal solution is by adding the right inductive bias. For the highly structured visual data, we study the best inductive bias we can add to prevent the shortcut during adaptation.

In addition to convolution, we also investigated the popular low-rank structure [24, 67]. We create visual prompts with low-rank structure (LVP) via singular value decomposition (SVD), and optimize the low-rank matrices using our algorithm (shown in Appendix 7.2). We compare the effectiveness of low-dimensional prompts and convolution on their ability to reverse natural corruptions. In Table 6 and Figure 3, for both LVP and CVP, increasing the prompt's rank leads to degraded performance in reverse the corruption to produce clean images, where the $\ell_2$ norm distance between real shifting and the approximated shifting increase.

|  | CIFAR-10-C Avg. Error | ImageNet-C mCE | ImageNet-R Error | ImageNet-S Error |
|---|---|---|---|---|
| Standard | 58.24 | 76.87 | 63.83 | 75.90 |
| SSL-VP | 57.95 (-0.29) | 76.74 (-0.13) | 68.86 (+5.03) | 75.93 (+0.03) |
| SSL-CVP | **52.37 (-5.87)** | **75.34 (-1.53)** | **63.06 (-0.77)** | **75.30 (-0.60)** |
| Sup-VP | **42.02 (-16.22)** | **73.54 (-3.33)** | **60.17 (-3.66)** | **73.30 (-2.60)** |
| Sup-CVP | 49.59 (-8.65) | 74.10 (-2.77) | 62.60 (-1.23) | 74.14(-1.76) |

Table 5: Analysis on the shortcut of SSL. We compare the performance of SSL and supervised learning on VP and CVP for four benchmarks

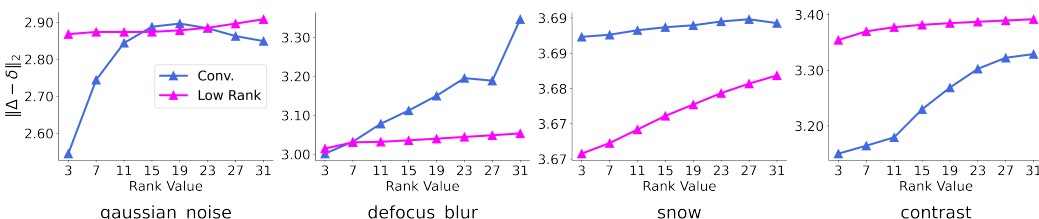

Figure 3: The effect of structured prompt for reversing corruption. For the y-axis, we show the $\ell_2$ distance between the ground-truth additive corruptions vector $\Delta$ and our prompt reversing vectors $\delta$, where a small distance indicates our reversal is successful. For the x-axis, we show the rank for the prompt matrices and the convolutional kernel. For a convolutional kernel with size $k \times k$, its rank is k. We find a small rank help reverse the noise better. For the same rank, convolution is better than the additive matrix. This demonstrates the effectiveness of our convolution prompt over other low-rank structures in reversing natural corruptions.

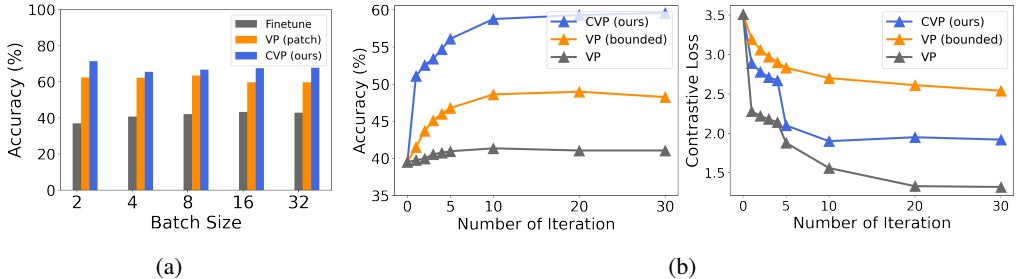

Figure 4: (a.) Analysis of the effect of batch size for different baselines on Cifar-10-C. We show the average performance on severity 1 for every corruption type. When batch size is small, CVP has a better performance on all corruption types, compared to fine-tune and two VP methods. (b.) We show the analysis for self-supervised loss v.s. performance on VP and CVP for gaussian corruption type in CIFAR-10-C. We setup the norm-bound for VP as 8/255 or unbounded. The loss curve demonstrates that when increasing the iter. number from 1 to 30, adapting with VP without norm bound hugely over-fits the self-supervised loss. The loss of VP largely reduces, yet the performance does not improve. On the other hand, our CVP has a lower risk of over-fitting as the loss curve can reduce smoothly with accuracy gradually increasing.

**CVP in Supervised Learning.** CVP is necessary for our test-time adaptation because SSL is a proxy for our final task. There are solutions where SSL is minimized, but the final task performs suboptimal. We investigate the necessity of using CVP when optimized for objectives other than SSL, such as the supervised loss, where the training and evaluation objective matches. Hence, in Table 5, we show the VP and CVP under the supervised learning setting (Sup-VP and Sup-CVP) and SSL setting (SSL-VP and SSL-CVP). For all experiments, we use the same optimization setting and parameter settings as Table 4. We apply the prompts on every batch for the supervised setting and update them with ground truth. Due to training on the ground-truth label, both VP and CVP achieve higher accuracy than SSL, VP performs higher than CVP. The results demonstrate that convolution inductive bias is not necessary when ground truth is available. However, without the label, the structured convolutional prompt is necessary for improving performance.

**Generalize CVP to other SSL tasks.** We further show the CVP can be generalized to other self-supervision tasks. In Table 7, we compare three SSL tasks, including contrastive learning, rotation prediction and masked autoencoder (MAE) [15]. We show our CVP is effective under all SSL tasks. On ImageNet-C, CVP improves robust accuracy for every severity level (from s1 to s5) when optimized for all three SSL tasks. We find contrastive learning can perform better than rotation prediction, and MAE has the best performance.

**The Effect of Batch Size and Number of Iteration for Adaptation.** In Figure 4a, we empirically demonstrate that the batch size affects the performance of different prompt methods. We set up the batch sizes from 2,4,8,16 to 32 and compared the accuracy of CVP with Finetune, VP (padding), and VP (patch). When batch size is set as 2, the performance of fine-tune is only 36.95%, which is worse than CVP (71.53%). When increasing the batch size to 4 and 8, the performance of fine-tune, VP slightly improves, yet still worse than CVP. Overall, CVP has an advantage in adaptation under the

|            | Standard | LVP_R3         |
| ---------- | -------- | -------------- |
| CIFAR-10-C | 58.24    | 54.86 (-3.38)  |
| ImageNet-C | 76.87    | 76.42 (-0.45)  |
| ImageNet-R | 63.83    | 63.57 (-0.26)  |
| ImageNet-S | 75.90    | 75.69 (-0.21)  |

|          | S1    | S2    | S3    | S4    | S5    |
| -------- | ----- | ----- | ----- | ----- | ----- |
| Standard | 58.82 | 48.25 | 38.65 | 27.15 | 17.57 |
| Contrast | 59.53 | 48.84 | 39.82 | 28.55 | 19.53 |
| Rotation | 58.97 | 48.30 | 38.88 | 27.50 | 17.99 |
| MAE      | 60.94 | 51.10 | 41.79 | 30.11 | 19.92 |

Table 6: Performance of low-rank visual prompt. We show the average error rate on four benchmarks. For CIFAR-10-C, we use WideResnet as the standard model. For ImageNet-C, R, and S, we use the ResNet50.

Table 7: Performance of CVP on three SSL tasks for ImageNet-C, including contrastive learning, rotation prediction, and MAE. We compare them with the Standard ResNet50 and show the averaged accuracy for 15 corruption types on every severity.

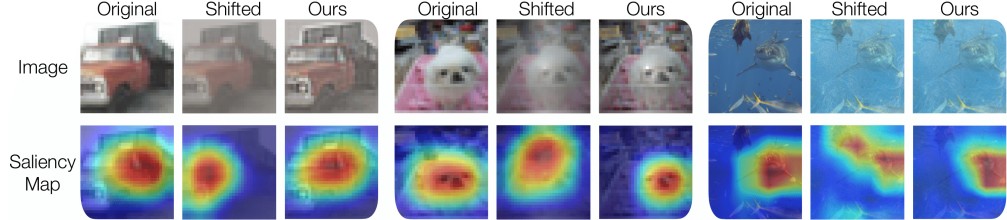

Figure 5: Visualization. From left to right we show three kinds of corruption types, including contrast, fog, and frost, on ImageNet examples. By applying our convolutional prompt on the corrupted images, our method can partially remove the corruptions and make the image easier to recognize. In addition, the saliency map calculated from Grad-Cam also shows that our approach instructs the model to look at a similar region as the original one. This highlights why our convolutional can adapt the input for robustness.

small batch setting. In Figure 4b, we show the performance on different numbers of iterations for adaption. We empirically found that when increasing the iterations, CVP has a lower risk to overfit on the self-supervised objective. More ablation studies, such as different prompt designs can be found in Appendix 7.7

**Visualization of Saliency Map** To better understand how CVP adapts to the corrupted inputs, we visualize the saliency map of different types of corruption. As Figure 5 shows, from left to right, the first row is the original, corrupted, and adapted samples; the second row shows their corresponding Grad-CAM with respect to the predicted labels. The red region in Grad-CAM highlights where the model focuses on target input. We empirically discover the heap map defocuses on the target object for corrupted samples. After CVP, the red region of the adapted sample's heap map is re-targeted on a similar part as the original image, demonstrating that the self-supervised visual prompts indeed improve the input adaptation and make the model refocus back on the correct areas. We provide more visualization in Appendix 7.10.

**Training Cost v.s. Different Kernel Size** In Table 8, we evaluate different kernel sizes for CVP and empirically discover that increasing the kernel to a proper size can improve the performance slightly. We choose one corruption-type impulse noise and show the results. When increasing the kernel size, the optimization cost increases. For impulse noise, kernel size 7*7 achieves the best robust accuracy, yet the optimization cost is much higher.

| Kernel Size | Accuracy (%) | Training Cost/Batch |
| ----------- | ------------ | ------------------- |
| 3*3         | 16.22        | 0.67s               |
| 5*5         | 16.3         | 0.68s               |
| 7*7         | **16.62**    | **1.24**s           |
| 13*13       | 16.61        | 1.28s               |
| 21*21       | 16.52        | 1.29s               |
| 25*25       | 15.4         | 1.32s               |

Table 8: Training Cost v.s. Different Kernel Size

**Training Time v.s. Number of Adapt Iteration** In Figure 4(b), we have shown the CVP trained under different adapt iterations v.s. their performance. When increasing the number of adapt iterations, the training time increases. The following Table 9 shows the result of CIFAR-10-C on gaussian noise type with severity 1. We compare the accuracy and per batch training time on several numbers of

adapt iters (from 0 to 20). We empirically found that CVP has a larger performance gain than VP while adapting with a few epochs (epoch number 1).

| # of Adapt Iters | 0 | 1 | 5 | 10 | 15 | 20 |
|---|---|---|---|---|---|---|
| Cost/Batch | 0.00s | 0.17s | 0.67s | 1.29s | 1,.92s | 2.57s |
| CVP Acc.(%) | 39.51 | 51.09 | 56.1 | 58.76 | 59.30 | 59.58 |

Table 9: Training Time v.s. Number of Adapt Iteration

**Does CVP Reverse Corrupted Images Back to Normal One?** We do the quantitative measurement on the distribution distance via Sliced Wasserstein Distance (SWD) and structural similarity index measure (SSIM). We measure the distance between two input distributions: source domain distribution and target domain distribution (before/after CVP adaptation). To calculate the distance between two input distributions via the Sliced Wasserstein Distance, we first obtain a group of marginal distributions from a high dimensional probability distribution via the linear projection, then calculate the $p$-Wasserstein Distance for those marginal distributions. Table 10 and Figure 6 shows the result of SWD on CIFAR-10-C with severity 1. On average, CVP achieves lower SWD after adaptation, which means the target distribution is closer to the source one after adaptation. The average SWD reduce by 0.7% after prompting. In Appendix Table 18 and Figure 9, we show the more detailed analysis.

| | SWD (scale: $10^2$) $\downarrow$ | | SSIM $\uparrow$ | |
|---|---|---|---|---|
| | before | after | before | after |
| Avg. Mean | 7.19 | **6.49** | 0.7539 | **0.7884** |
| Avg. Std | 4.05 | **2.79** | 0.1294 | **0.7260** |

Table 10: Results of SWD and SSIM on CIFAR-10-C (Severity 1).

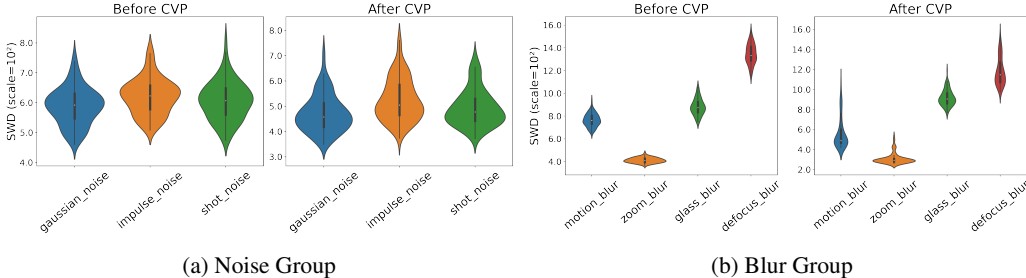

(a) Noise Group  (b) Blur Group

Figure 6: Violin Plot of SWD for different corruption groups on CIFAR-10-C (severity 1). The left figure of each subplot shows the SWD before adapting, and the right shows the SWD after CVP adaptation. CVP brings the corrupted image distribution back to the clean image distribution.

## 6   Conclusion

Self-supervised convolutional visual prompt (CVP) is a novel method for test-time adaptation of OOD samples. In contrast to prior works training the visual prompts with the label, CVP is label-free and lightweight. It reduces the trainable parameters to less than 1% parameters of previous visual prompts and avoids the risk of overfitting when adapting for self-supervised objectives at test time. Results on five state-of-the-art benchmarks show that CVP improves model robustness by 5.87% and complements existing weight-adaptation methods. Extensive ablation studies suggest that distribution shifts are actually structured; therefore, CVP can capture the structures better than VP during the adaptation, which provides new insight into the frontier of visual prompting techniques and test-time adaptation. Future work includes interpreting convolutional prompts and prompting with multi-modality in large-scale foundation models.

## Acknowledgement

This research is supported in part by a GE/DARPA grant, a CAIT grant, and gifts from Google and Accenture. We thank suggestions from Yi Zhang, Yow-Kuan Lin, and Raphael Jedidiah Sofaer.

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

# 7 Appendix

## 7.1 Flow of CVP

We illustrate the whole flow of CVP in Figure 7. There are two phases in our proposed method: (1.) Offline one-time training on SSL model: The SSL is a simple MLP model, where it takes the logit features from pre-trained backbone as input and trains the model with contrastive learning task. (2.) CVP adaptation for any OOD benchmark during test time: We leverage the SSL loss from well-trained SSL model to optimize the convolutional kernel prompts for every upcoming OOD sample.

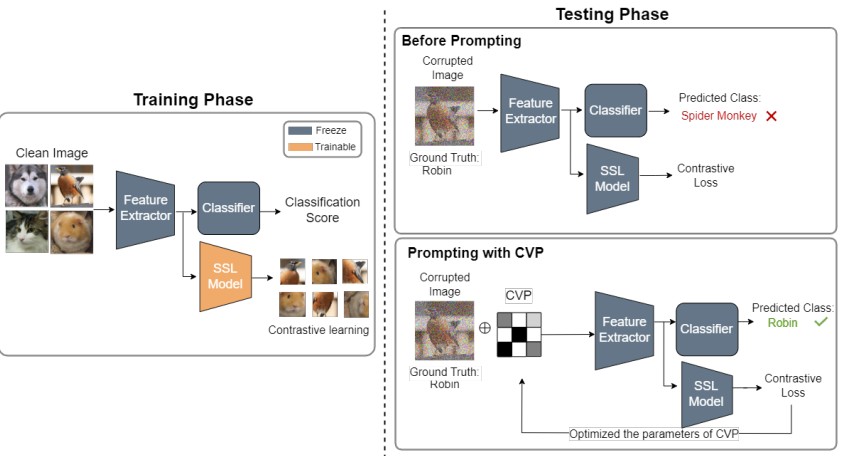

Figure 7

## 7.2 Algorithm

## 7.3 Baseline Details

Here, we show the detail of the baselines that we compare with.

● **Standard**: The baseline uses pre-trained model without adaptation. For CIFAR-10-C, the standard is trained with 50000 clean CIFAR-10 train dataset on WideResNet18 and ResNet. For ImageNet1K-C, the standard is trained with ∼1.2M clean ImageNet train dataset on ResNet50.

● **SVP**: The self-supervised visual prompts attempt to reverse the adversarial attacks by modifying the input pixels with $\ell_p$-norm perturbations, where the perturbations are optimized via contrastive loss [40]. For the patch setting, we setup the shape of VP as 32*32*3 for CIFAR-C and 224*224*3 for all ImageNet OOD datasets. For the padding setting, we set the padding size as 1 for CIFAR-10-C and 15 for ImageNet OOD dataset. Take CIFAR data as example, we first initialize a mask with all zeros value with the shape 30*30*3 and set the pad value as 1 with padding size 1 so that the mask after padding is as the same shape of CIFAR data (32*32*3). Then, we multiply the mask with the VP to preserve only the VP located at the position we just pad with 1 value. We can further optimize the VP with mask by adding it with the corrupted samples.

● **BN[55]**: The model adaptation method aims to adjust the BN statistics for every input batch during the test-time. It requires to adapt with single corruption type in every batch.

● **TTT [59]**: The test-time training trains the model with an auxiliary SSL rotation task and leverages the rotation loss for model adaptation during the testing time. In TTT method, instead of adapting the whole model, they only adapt the last few layers of the model and freeze the parameters in the front layers.

● **MEMO**: The model adaptation method proposed in [71] alters a single data point with different augmentations (ie., rotation, cropping, and color jitter,...etc), and the model parameters are adapted by minimizing the entropy of the model's marginal output distribution across those augmented samples.

---

**Algorithm 1:** Convolutional Visual Prompt

---

**Input:** Pretrained classifier $\mathcal{F}(\cdot)$, OOD images $x$, Self-supervised objective function $\mathcal{L}_s(\cdot)$, Convolutional operator $Conv(\cdot)$, Convolutional kernel $k$, Learning rate $\eta$, Number of iteration $\mathcal{T}$

**Output:** Class prediction $\hat{y}$ for adapted sample of $x$

**1 Inference**
**2 # Initialize the kernel parameters**
**3** $k^0 \sim \mathcal{U}\{(\alpha, \beta)$
**4 # Calculate initial SSL loss**
**5** $loss^0 = \mathcal{L}_s(x)$
**6 for** $t \in \{1, ..., T\}$ **do**
**7**    **# Generate adapted samples**
**8**    $x^t = x + \lambda * Conv(x, k^t)$
**9**    **# Calculate SSL loss with adapted samples**
**10**    $loss^t = \mathcal{L}_s(x^t)$
**11**    **# Update kernel parameters**
**12**    $k^{t+1} = k^t + \eta \frac{\partial loss^t}{\partial k^t}$
**13 # Get optimal kernel parameters**
**14** $k^\star \leftarrow k^T$
**15 if** $loss^T > loss^0$ **then**
**16**    **# Use the initial kernel parameters**
**17**    $k^\star \leftarrow k^0$ ;
**18 # Get final adapted samples**
**19** $x^\star = x + \lambda * Conv(x, k^\star)$
**20 return** $\hat{y} \leftarrow \mathcal{F}(x^\star)$

---

---

**Algorithm 2:** Low-Rank Visual Prompt

---

**Input:** Pretrained classifier $\mathcal{F}(\cdot)$, OOD images $X$, Self-supervised objective function $\mathcal{L}_s(\cdot)$, Rank Number $r$, Learning rate $\alpha$, Number of iteration $\mathcal{K}$

**Output:** Class prediction $\hat{y}$ for adapted sample of $x$

**1 Inference**
**2 # Initialize the matrices parameters**
**3** SVD on input $X(N * C * H * W) \rightarrow$ get n pairs of initial $U\Sigma V^T$ (3 channel)
**4 for** $t \in \{1, ..., K\}$ **do**
**5**    **# Get N inverse matrices**
**6**    $M_t \leftarrow U_t \times \Sigma_t \times V_t^T$
**7**    **# Apply low-rank SVD on** $M_t$ **with rank** $r$
**8**    $U_t', \Sigma_t', V_t^{T'} = SVD_{low-rank}(M_t)$
**9**    **# Generate adapted samples with low-rank inverse matrices**
**10**    $M_t' = U_t' \times \Sigma_t' \times V_t^{T'}$
**11**    $X_t' = X_t + M_t'$
**12**    **# Calculate SSL loss with adapted samples**
**13**    $loss^t = \mathcal{L}_s(X_t')$
**14**    **# Update the three matrices**
**15**    $U_{t+1} = U_t + \alpha \frac{\partial loss^t}{\partial U_t}, \quad \Sigma_{t+1} = \Sigma_t + \alpha \frac{\partial loss^t}{\partial \Sigma_t}, \quad V_{t+1}^T = V_t^T + \alpha \frac{\partial loss^t}{\partial V_t^T}$
**16 # Get optimal low rank matrix**
**17** $M^\star \leftarrow U_K \times \Sigma_K \times V_K^T$
**18 # Get final adapted samples**
**19** $X^\star = X + M^\star$
**20 return** $\hat{y} \leftarrow \mathcal{F}(X^\star)$

---

• **TENT [63]**: The method adapts the model by minimizing the conditional entropy on batches. In our experiment, we evaluate TENT in *episodic* mode, which means the model parameter is reset to the initial state after every batch adaptation.

### 7.4 Implementation details

For the training part of SSL model, we set the training parameters with batch size as 64, training epoch as 200, and the learning rate (*lr*) as 0.001. The *lr* is decayed with a cosine annealing for each batch [37]. The transformations for contrastive learning are predefined. We augment the inputs with random resize crop, random flip, and random rotation in degree [-90, 90] for positive/negative pairs generation in every batch. The number of transformations for one sample is set as 3.

For the test-time adaptation part, we set the range of parameter $\delta$ for VP. For the $\ell_2$-norm perturbations, the $\epsilon$ is [-8/255,8/255] and the step size is 2/255. We set the iteration number $i$ either as 1 or 5, which means each component has 1 or 5 steps during PGD. For the update iterations, as the Table 17. show a larger number of iterations has better performance. However, the training cost becomes higher.

• **The choice of hyper-parameter setting on $\lambda$ ranges and kernel**

For the $\lambda$ parameter, it controls the magnitude of convolved output when combined with the residual input. We set the range to be [0.5, 3] and run test-time optimization to automatically find the optimal solution, which does not require a validation set.

We use 3x3 for cifar-10 and 5x5 for ImageNet. In general, small kernel size is used to avoid overfitting. We increase the kernel size for large images, such as ImageNet.

When adapting, the kernel should be initialized with fix/random initialization. We use a sharpness kernel as an initial point for the fixed initialization setting, which is a n*n matrix. It starts from specific values, which can control the edge enhancement effect by extracting the frequency information of inputs with a high pass filter. When the kernel size is 3, we set up the sharpness kernel as [[0,-1, 0], [-1, 5, -1], [0, -1, 0]] Similar to the convolutional kernel prompt, sharpness transforms an image using a convolutional 2D filter. It simultaneously enlarges the high-frequency part of images and then filters the low frequencies part.

|  | CIFAR-10-C | ImageNet-C,R,S,A |
|---|---|---|
| Kernel Size | 3*3 | 3*3 / 5*5 |
| $\lambda$ | [0.5, 1] | [0.5, 3] |
| Update iters. | 1, 5 (default), and 10 | |
| Initialization | fixed / random | |

Table 11: parameter setting

• **Number of Trainable Parameters**: We compare the trainable parameters v.s. accuracy for different prompting methods. As Figure 8 shows, CVP contains less than 0.2% number of trainable parameters, compared to VP(patch).

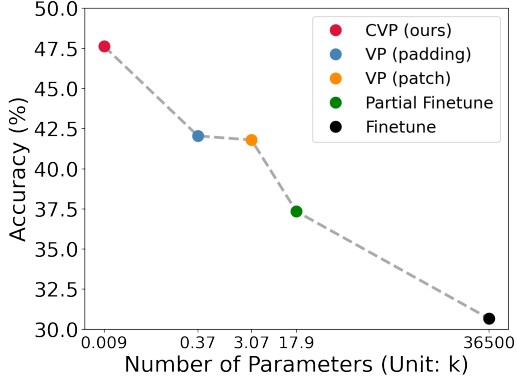

Figure 8

## 7.5 More Evaluation

• We show more detailed results for CIFAR-10-C in Table 12 and 13.

| | Standard | BN | Finetune | VP | | CVP | | | |
|---|---|---|---|---|---|---|---|---|---|
| | | | | patch | append | fixed (3*3) w/o update | fixed (3*3) w/ update | rand (3*3) w/o update | rand (3*3) w/ update |
| Gaussian Noise | 19.90 | 24.51 | 19.26 | 20.07 | 19.92 | 23.11 | 23.50 | 24.59 | **26.27** |
| Shot Noise | 20.37 | 24.25 | 19.43 | 20.56 | 20.40 | 22.95 | 23.32 | 24.29 | **25.26** |
| Impulse Noise | 27.44 | 26.14 | 19.19 | 27.54 | 27.44 | 30.76 | 30.98 | 31.13 | **31.08** |
| Defocus Blur | 12.90 | 14.54 | 13.39 | 13.59 | 13.18 | 16.81 | 17.41 | 18.85 | **20.03** |
| Motion Blur | 23.26 | 19.95 | 27.67 | 30.50 | 23.29 | 29.21 | 30.03 | 30.37 | **31.89** |
| Glass Blur | 25.97 | 33.30 | 18.61 | 19.70 | 26.01 | 36.14 | 37.54 | 37.42 | **40.51** |
| Zoom Blur | 71.08 | 50.46 | 42.42 | 71.07 | 71.06 | 86.76 | 87.65 | 85.76 | **88.19** |
| Brightness | 89.38 | 71.47 | 61.03 | 89.39 | **89.40** | 89.37 | 89.39 | 89.25 | 89.31 |
| Snow | 71.21 | 49.73 | 39.95 | 71.52 | 71.22 | 71.23 | 71.44 | 71.17 | **71.52** |
| Frost | 74.83 | 58.40 | 46.35 | **74.93** | 74.84 | 74.79 | 74.81 | 74.69 | 74.90 |
| Fog | 45.69 | 42.45 | 32.96 | 46.52 | 45.76 | 50.08 | 51.42 | 49.64 | **51.65** |
| Contrast | 58.36 | 49.87 | 39.22 | 57.95 | 58.47 | 67.74 | 68.99 | 69.05 | **70.21** |
| Elastic Transform | 17.54 | **24.01** | 19.95 | 17.72 | 17.56 | 17.39 | 17.62 | 18.07 | 19.66 |
| Pixelate | 23.45 | **39.80** | 34.26 | 24.10 | 23.50 | 25.91 | 26.47 | 28.39 | 30.58 |
| Jpeg Compression | 45.06 | 31.37 | 26.42 | **45.65** | 45.17 | 43.99 | 44.44 | 40.74 | 43.43 |
| Avg. Acc. | 41.76 | 37.35 | 30.67 | 42.05 | 41.82 | 45.75 | 46.33 | 46.23 | **47.63** |
| Avg. Error | 58.24 | 62.65 | 69.33 | 57.95 | 58.18 | 54.25 | 53.67 | 53.77 | **52.37** |
| Avg Diff. | - | 4.41 | 11.09 | -0.29 | -0.06 | -3.99 | -4.57 | -4.46 | **-5.87** |

Table 12: Comparison of the different prompting methods with CVP for every CIFAR-10-C corruption type. The Standard model is WideResNet18. Number in bold shows the best performance.

| Severity / Method | Standard | BN | Finetune | VP | | CVP | | | |
|---|---|---|---|---|---|---|---|---|---|
| | | | | Patch | Append | fixed (3*3) w/o update | fixed (3*3) w/ update | rand. (3*3) w/o update | rand. (3*3) w/ update |
| S1 | 59.68 | 52.48 | 43.28 | 59.94 | 59.76 | 65.17 | 66.00 | 65.98 | **68.07** |
| S2 | 47.88 | 43.18 | 34.72 | 48.26 | 47.94 | 52.73 | 53.42 | 53.41 | **54.73** |
| S3 | 40.31 | 36.44 | 29.08 | 40.67 | 40.32 | 44.22 | 44.87 | 44.51 | **45.96** |
| S4 | 32.75 | 29.49 | 24.66 | 32.94 | 32.75 | 36.20 | 36.74 | 36.56 | **37.79** |
| S5 | 28.20 | 25.16 | 21.63 | 28.46 | 28.25 | 30.42 | 30.64 | 30.68 | **31.61** |
| Avg. Acc. | 41.76 | 37.35 | 30.67 | 42.05 | 41.82 | 45.75 | 46.33 | 46.23 | **47.63** |
| Avg. Error | 58.24 | 62.65 | 69.33 | 57.95 | 58.18 | 54.25 | 53.67 | 53.77 | **52.37** |
| Avg Diff. | - | 4.41 | 11.09 | -0.29 | -0.06 | -3.99 | -4.57 | -4.46 | **-5.87** |

Table 13: Comparison of the different adaptation baslines with CVP for every severity on CIFAR-10-C. The Standard model is WideResNet18. Number in bold shows the best performance.

• More evaluation on ViT-Base model architecture. As Table 14 shows, we evaluate on 15 types of corruption under severity level 1 for CIFAR-10-C. The test accuracy of ViT-Base clean CIFAR-10 is 96.42%. Compared to the accuracy (%) before adaptation (standard) and using VP to do the adaptation, the CVP achieves better performance by up to 2 points.

| | ViT-Base | VP | CVP |
|---|---|---|---|
| Gaussian Noise | 47.64 | 49.71 | 50.22 |
| Shot Noise | 41.92 | 45.67 | 46.03 |
| Impulse Noise | 80.77 | 80.39 | 82.13 |
| Glass Blur | 48.49 | 47.77 | 49.12 |
| Defocus Blur | 27.91 | 27.37 | 28.90 |
| Zoom Blur | 83.71 | 83.23 | 85.63 |
| Motion Blur | 58.34 | 56.22 | 58.87 |
| Brightness | 94.40 | 93.10 | 94.59 |
| Snow | 89.37 | 88.23 | 89.41 |
| Frost | 90.13 | 92.19 | 92.32 |
| Fog | 74.75 | 73.60 | 75.63 |
| Contrast | 90.14 | 91.34 | 92.34 |
| Pixelate | 66.06 | 66.37 | 63.53 |
| Jpeg Compression | 62.53 | 61.55 | 63.53 |
| Elastic Transform | 40.68 | 40.52 | 41.52 |
| Averaged Acc. | 66.45 | 66.48 | **67.77** |

Table 14: Performance on ViT-Base model.

• We show the detailed results for each corruption on ImageNet-C dataset in Table 15.

• In Table 16, we further compare with TTT [59], which is a test-time method with CVP. We combine our CVP on top of TTT, where the model weight will first adapt based on the rotation loss, and then the input will adapt by convolutional visual prompt.

| | Standard | Finetune | BN Adapt | VP | | CVP | | | |
|---|---|---|---|---|---|---|---|---|---|
| | | | | patch | append | fixed 3*3 | rand 3*3 | fixed 5*5 | rand 5*5 |
| Gaussian Noise | 80.00 | 78.85 | 79.43 | 79.44 | 79.99 | 78.49 | **78.16** | 78.47 | 78.75 |
| Shot Noise | 82.00 | 80.80 | 81.57 | 81.56 | 81.97 | 80.45 | **80.00** | 80.10 | 80.81 |
| Impulse Noise | 83.00 | 81.80 | 82.72 | 82.72 | 83.00 | 80.80 | **79.82** | 80.88 | 81.40 |
| Defocus Blur | 73.58 | 75.49 | 75.32 | 77.27 | **73.56** | 74.13 | 73.73 | 74.29 | 75.10 |
| Motion Blur | 90.95 | 79.85 | 92.38 | 80.18 | **77.96** | 89.99 | 89.31 | 89.14 | 88.60 |
| Glass Blur | 76.32 | 87.16 | 76.86 | 90.41 | 79.96 | 75.98 | 75.47 | 75.99 | **75.45** |
| Zoom Blur | 80.00 | 79.84 | 80.52 | 82.07 | 88.98 | 79.87 | 79.67 | 79.72 | **79.45** |
| Snow | 43.86 | 45.10 | 45.82 | 47.88 | 47.95 | **44.24** | 44.27 | 44.60 | 44.91 |
| Frost | 79.88 | 81.04 | 82.22 | 83.78 | **74.99** | 80.12 | 80.19 | 79.69 | 80.05 |
| Fog | 74.38 | 75.74 | 76.80 | 78.06 | **64.38** | 74.53 | 74.91 | 74.36 | 74.85 |
| Brightness | 78.25 | 79.90 | 81.23 | 84.99 | 88.07 | 78.78 | **78.49** | 78.83 | 78.91 |
| Contrast | 71.00 | 73.27 | 74.14 | 76.59 | 70.98 | 71.46 | **70.83** | 71.31 | 71.79 |
| Elastic Transform | 87.58 | 87.96 | 88.89 | 96.50 | 87.62 | 87.93 | 87.85 | **87.42** | 88.15 |
| Pixelate | 74.72 | 74.22 | 75.75 | 78.30 | 74.68 | 67.05 | **63.98** | 67.29 | 64.15 |
| Jpeg Compression | 77.00 | 75.47 | 74.85 | 81.29 | 76.99 | 74.41 | **73.46** | 74.05 | 74.26 |
| mCE | 76.83 | 77.10 | 77.90 | 80.07 | 76.74 | 75.88 | **75.34** | 75.74 | 75.77 |
| Diff. | | 0.27 | 1.07 | 3.24 | -0.09 | -0.95 | **-1.49** | -1.09 | -1.06 |

Table 15: ImageNet-C results. Number in bold shows the lowest mCE.

| | WideResNet18 Avg. Error (%) |
|---|---|
| **Standard** | 58.24 |
| VP (patch) | 57.94 (-0.3) |
| CVP (rand. w/ update) | **52.37 (-5.87)** |
| TTT [63] | 52.92 (-5.32) |
| TTT + CVP | 53.07 (-5.17) |

Table 16: TTT [59] result for CIFAR-10-C

• **Generalize to Cutout-and-Paste samples** To justify that our method can be generalized to non-structured OOD, we do more experiments on other types of OOD samples, such as the Cutout-and-Paste samples. Here, we launch the experiment on the **Waterbirds** dataset, which is constructed by cropping out birds from images with "water" backgrounds in the Caltech-UCSD Birds-200-2011 (CUB) dataset [61] and transferring them onto backgrounds from the Places dataset [72]. We follow the GitHub repo[*] and choose the "Forest" as our new background to generate the samples. The training of SSL is based on the pre-trained ResNet34 backbone model for the original CUB dataset. The original CUB (200 classes) accuracy for the backbone ResNet34 is 75.34%. We compare our CVP with self-supervised VP and demonstrate that CVP is more effective on the Cutouted-CUB dataset. The following table shows our results. Our CVP improves the result upon Standard by 1.61 points and VP by 1.3 points.

| **Cutouted-CUB (200)** | **Before Adapt** | **VP (patch)** | **CVP** |
|---|---|---|---|
| Accuracy (%) | 62.03 | 62.32 | **63.64** |
| contrastive loss (Avg.) | 2.78 | 2.71 | **2.52** |

Table 17: Performance on the Cutouted-CUB

---

[*]WaterBirds Dataset https://github.com/kohpangwei/group_DRO

## 7.6 Distance measurement with SWD and SSIM

As the main paper mentioned, we measure the SWD and SSID on two input distributions: source domain distribution and target domain distribution (before/after adaptation). In Table 18 and Figure 9, we show the detailed result of SWD on every corruption type in CIFAR-10-C under severity 1. On average, CVP achieves lower SWD after adaptation, which means the target distribution is closer to the source one after adaptation. The average SWD reduced by 0.7% after prompting.

|  | SWD (scale: $10^2$) $\downarrow$ |  | SSIM $\uparrow$ |  |
|---|---|---|---|---|
|  | before | after | before | after |
| Gaussian Noise | 5.90 | **4.71** | 0.7242 | **0.7849** |
| Shot Noise | 6.08 | **4.93** | 0.7124 | **0.7676** |
| Impulse Noise | 6.23 | **5.26** | 0.7463 | **0.7764** |
| Glass Blur | 8.85 | 9.19 | 0.5873 | 0.5865 |
| Defocus Blur | 13.52 | **11.82** | 0.6031 | 0.6013 |
| Zoom Blur | 4.13 | **3.09** | 0.8726 | 0.8703 |
| Motion Blur | 7.68 | **5.57** | 0.6491 | 0.6459 |
| Brightness | 2.48 | 3.94 | 0.9702 | 0.9692 |
| Snow | 5.18 | 6.07 | 0.8258 | **0.8275** |
| Frost | 7.61 | 7.72 | 0.8025 | 0.8012 |
| Fog | 13.49 | **9.99** | 0.5840 | 0.5785 |
| Contrast | 15.39 | **11.09** | 0.7049 | 0.6997 |
| Pixelate | 3.09 | 4.56 | 0.8603 | **0.8669** |
| Jpeg Compression | 2.58 | 3.65 | 0.8681 | **0.8710** |
| Elastic Transform | 5.62 | 5.75 | 0.5272 | **0.5789** |
| Avg. Mean | 7.19 | **6.49** | 0.7539 | **0.7884** |
| Avg. Std | 4.05 | **2.79** | 0.1294 | **0.7260** |

Table 18: Results of Sliced Wasserstein Distance and Structural Similarity Index Measure on CIFAR-10-C (Severity 1).

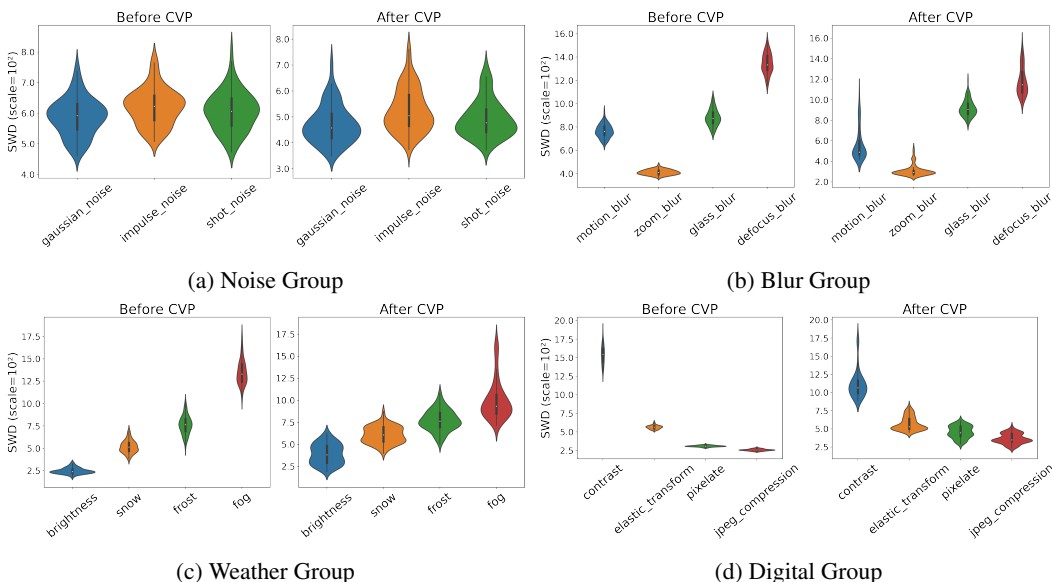

(a) Noise Group

(b) Blur Group

(c) Weather Group

(d) Digital Group

Figure 9: Violin Plot of SWD for different corruption groups on CIFAR-10-C. The left figure of each subplot shows the SWD before adapting, and the right shows the SWD after adaptation

• **Distribution changes after applying the proposed CVP**

In the main paper, Figure 2, we show the distribution changes in different corruption types and severity. Here, in Table 19, we show the distribution shifts after applying our CVP by calculating the average loss. In general, the distribution moves back to the original distribution. We show the SSL average loss (before adapt/after CVP adapt) for four corruption types on severity 1,3,5 for CIFAR10-C. The average SSL loss for the original CIFAR10 is 1.26. For every corruption we show here, the average SSL loss after adaptation is lower than the loss before adaptation.

| severity | s1 | s3 | s5 |
| --- | --- | --- | --- |
| | Before /After | Before /After | Before /After |
| Gaussian noise | 1.9 / 1.6 | 2.5 / 2.1 | 3.3 / 2.6 |
| Defocus blur | 3.2 / 2.9 | 3.4 / 2.8 | 3.7 / 3.1 |
| Snow | 3.1 / 3.0 | 3.8 / 3.3 | 3.9 / 3.5 |
| Contrast | 2.7 / 2.3 | 2.9 / 2.4 | 3.6 / 3.3 |

Table 19: Distribution changes on different corruption types.

## 7.7 The Effect of Different Prompt Designs

We do analysis on different prompting methods, including original visual prompts with different norm-bound ($\ell_2$, $\ell_\infty$), convolutional prompts, and their combinations ($\ell_2 + conv.$, $\ell_\infty + conv.$). We show the error rate on different numbers of adapt iters for every prompting method from 0, 1, 5, to 10. To compare the results, we set up other parameters such as the epsilon $\epsilon$ as $8/255$ for $\ell_\infty$, 1 for $\ell_2$. As Figure 10 shows the error rate for different prompting methods, the convolutional prompt $conv.$ and its combination with $\ell_2$ reduce the error rate, and the former one reduces more from 40.32% to 36.08% when increasing the adapt iters. However, other prompting methods increase the error rate after prompting. To understand the risk of over-fitting for different prompting methods, Figure 4b shows the SSL loss curve v.s. performance on different prompting methods.

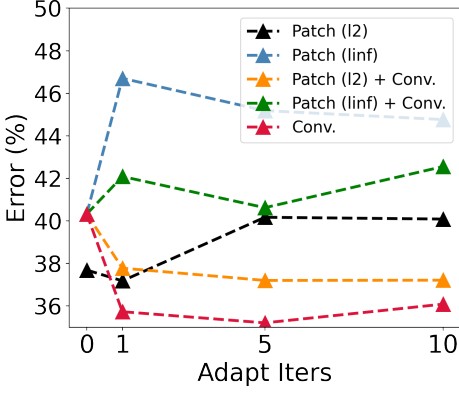

Figure 10

## 7.8 Analysis on CVP and low-rank prompts

In Table 20, we show the detailed results of low-rank prompt (LVP) on different severity (from 1 to 5) for CIFAR-10-C. We set up the same rank as 3 for LVP and CVP. Our results show that the CVP is more effective than LVP when reversing natural corruption. In Figure 11, we further plot the averaged contrastive loss on different rank sizes for both LVP and CVP. On every corruption type, while increasing the rank size from 3 to 31, the loss curves of LVP consistently drop, which demonstrates the LVP is much more easier to overfit the contrastive loss.

| Severity / Method | Standard | LVP | CVP |
|---|---|---|---|
| s1 | 40.32 | 37.05 | 31.93 |
| s2 | 52.12 | 48.83 | 45.27 |
| s3 | 59.69 | 56.05 | 54.04 |
| s4 | 67.25 | 63.42 | 62.21 |
| s5 | 71.80 | 68.89 | 68.39 |
| Avg. Error | 58.24 | 54.85 | **52.37** |
| Diff. | | -3.39 | **-5.87** |

Table 20

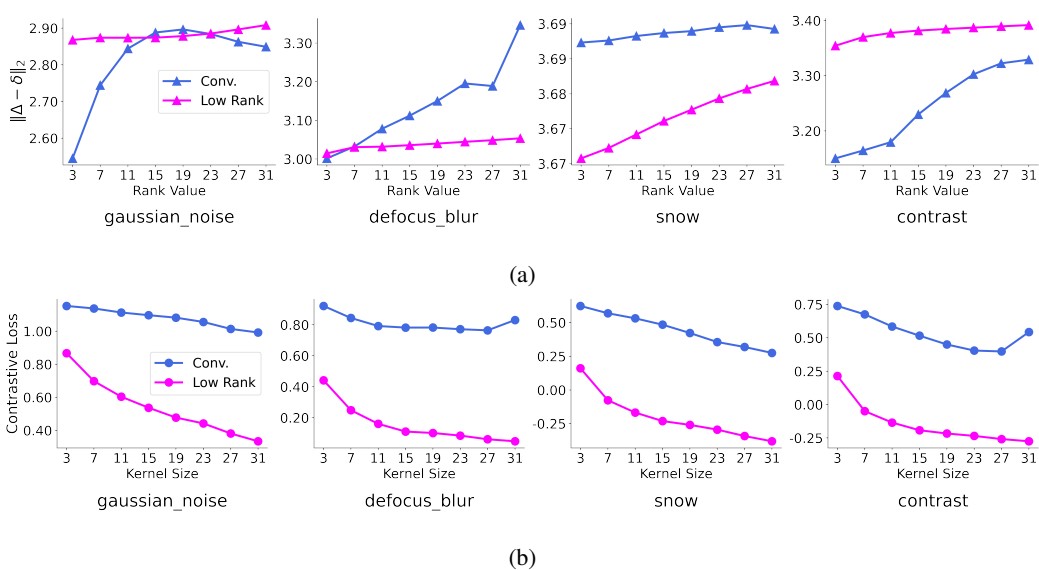

Figure 11: In subfigure a., We show the $\ell_2$ distance between the ground-truth additive corruptions vector $\Delta$ and our prompt reversing vectors $\delta$. Subfigure b shows the averaged contrastive loss of low-rank prompt and CVP on different rank sizes. For every corruption type, the loss curves of LVP consistently drop when increasing the rank to 31 while the $\ell_2$ distances increase, demonstrating how LVP is more prone to overfitting the contrastive loss.

### 7.9  t-SNE analysis on different adaptation methods.

In addition to performing analysis on single sample, we further conduct the t-SNE visualization for whole sample distribution on different baseline. For each type of corruption data, we extract the 1-dimensional logit features in the last layer of model and calculate the distance between them with respect to the predicted class labels. We compare our method with standard, MEMO, and MEMO + Ours. As Figure 12 shows, the original feature embedding shows low separability between different classes. On the other hand, our approach clearly discriminates the embedding feature, which demonstrates its robustness against distribution shifts.

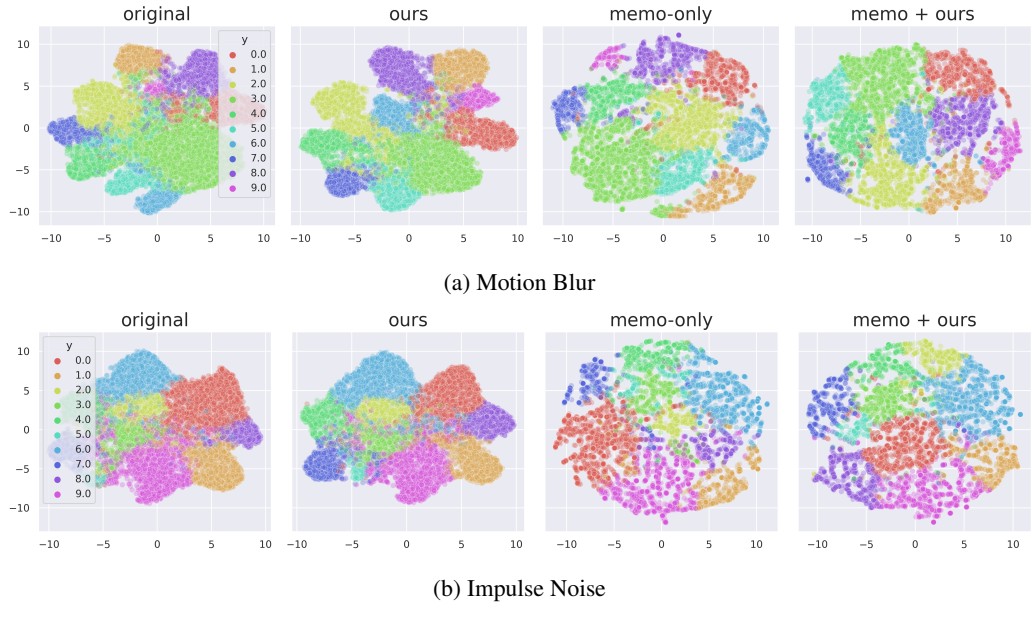

(a) Motion Blur

(b) Impulse Noise

Figure 12

### 7.10  Saliency map analysis on different corruption types

To better understand how self-supervised visual prompts adapt to the corrupted inputs, we visualize the saliency map of different types of corruption. As Figure 13 shows, from left to right, the first row is the original, corrupted, and adapted samples; the second row shows their corresponding Grad-CAM with respect to the predicted labels. The red region in Grad-CAM shows where the model focuses on for target input. We empirically discover the heap map defocus on the target object for corrupted samples. However, after prompting, the red region of the adapted sample's heap map is re-target on the similar region as original image, which demonstrates that the self-supervised visual prompts indeed improve the input adaptation and make the model refocus back on the correct regions.

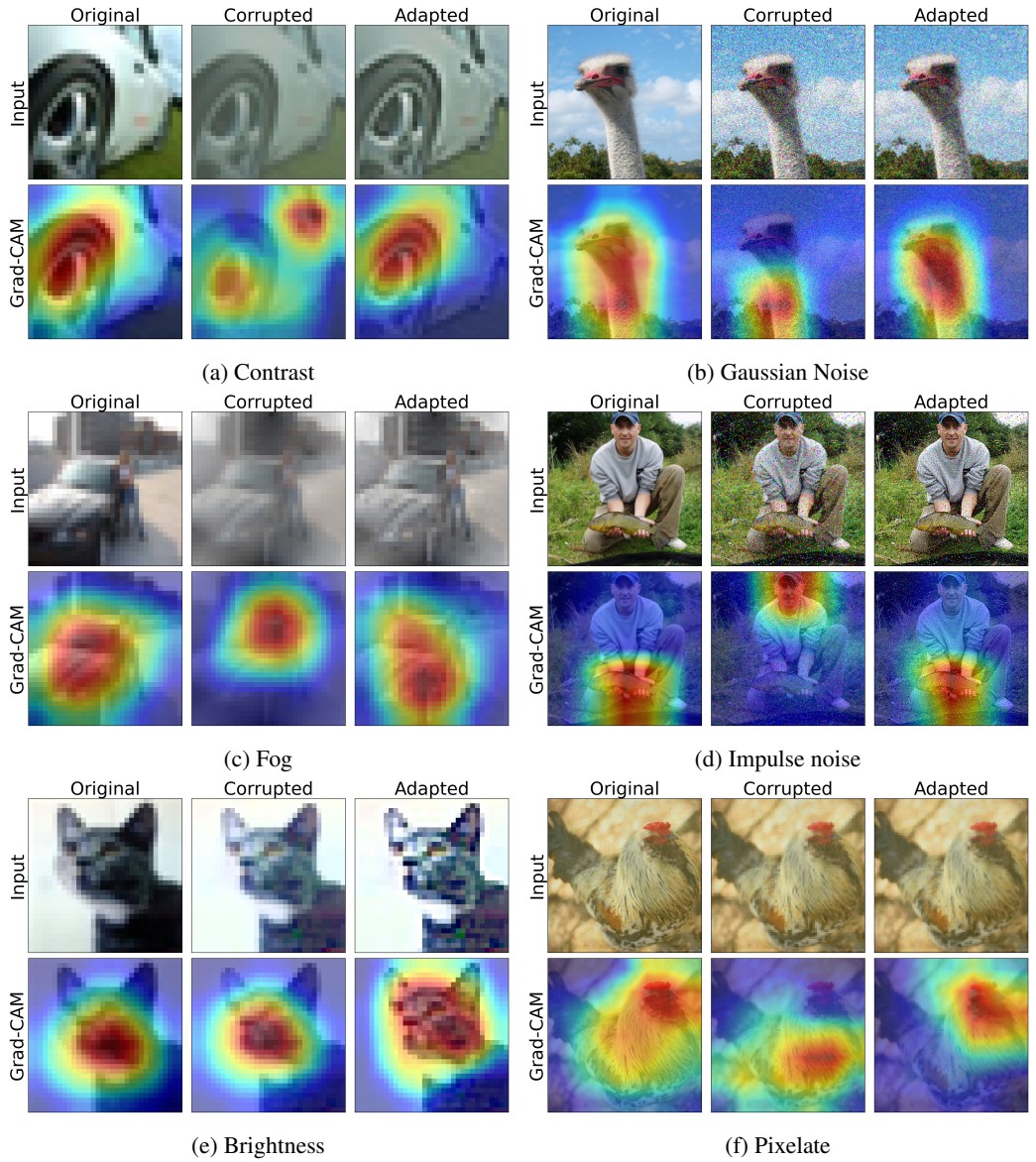

Figure 13: Grad-CAM analysis on different types of corruption.

