# OpenReview forum: "Convolutional Visual Prompt for Robust Visual Perception"
_NeurIPS.cc/2023/Conference — NeurIPS 2023 poster_

### Official Review · Reviewer_QKSZ · 2023-06-22

**Soundness:** 2 fair
**Presentation:** 2 fair
**Contribution:** 2 fair
**Rating:** 3
**Confidence:** 4

**Summary:**

The paper addresses test time adaptation by employing learnable convolution operation named CVP on out-of-distribution (OOD) images. The motivation is to take advantage of the inductive bias imparted by convolution operations with the added advantage of learning fewer parameters compared to existing approaches. CVP in conjunction with other sota approaches improve the performance on benchmark datasets.

**Strengths:**

1. Using small convolution operations at the start for OOD test time adaptation to take advantage of inherent inductive bias seems to be new.
2. The combination of selfsupervision and convolution makes it both parameter as well as label efficient.
3. Main and ablation experiments are largescale

**Weaknesses:**

1. Eqn. (1): When $y_{i,j}^s$ is 0, $i$ and $j$ are from different samples (negative pair). So, in the loss calculation negative pairs are not considered. Then how is it a contrastive formulation?
2. Line 111: It is said that $\mathcal{X}_s$ is the source domain data for training. If source domain data is used then it is not a test time adaptation framework and comparing it with such works would not be fair.
3. Line 150: One of the important assumptions made in this paper tells the distribution shift in images is often visually structured. Why is this assumption right? What tells that the distribution shift is 'visually structured'?
4. Line 180+: When describing the proposed model, it is told that the logits before the fully connected layer of the backbone model are extracted for training the SSL model. This is making me confused. Does it mean the conv operation is applied on the logits and not on the image? I think having a figure describing the proposed approach would have been useful here. I don’t see such a figure and thus I have to presume things. Also, it is said that a MLP gets trained in addition. Is it used during inference? Also, is this taken into account in the learnable parameter count, especially when compared with other related approaches like TENT or other traditional prompting methods? In this regard, the figure 6 in the supplementary is incomplete. What is the unit in x-axis? Millions, billions, K?
5. Line 184: When extending the proposed approach to CLIP, why only vision encoder is used for prompting? Why not text encoder only? Is there any ablation that is done?
6. Section 4.1: FT and PFT Baseline: Why is this restoring to initial weights required? Is it standard in literature? Any reference?
7. Line 206: What is a sharpness kernel?
8. Section 4.2, CVP Complements Other Test-Time Adaptation Methods: More details are required on what is meant by 'combining' CVP with these methods. Also, the results of this experiment (Table 4) are not convincing. Improvement is very marginal compared to TENT, in all three imagenet variants. Compared to standalone CVP, CVP+x provides huge error rate drop - then this means the major workforce behind the performance is x and not CVP.
9. Ablation study – Low rank structure: More details are needed on how the low rank prompts are created? Are they directly applied on the input? According to the original LoRA paper [24], low rank adaptation is shown to be effective in transformer architectures. How is it adapted to convolutional architectures? As two low rank matrices are involved (I'm supposing so, as details are missing), this evidently involves more learnable parameters and thus susceptible to overfitting. So, the comparison to conv kernels is not fair. The corresponding figure in Appendix (Fig. 9), does not have any descriptive captions. It’s better not to give such figures if everything about the figure has to be inferred by the reader.
10. Line 266-271 and Table 7: What is 'standard' SSL task? Is it employing eqn. (1)? Its difference with 'contrastive' variation needs to be detailed. While rotation or MAE were shown to perform worse compared to the standard SSL task (in Table 7), there is no detail of these two other SSL approaches. How many variations of rotation prediction were used? How much percentage of the images were masked in MAE? Is there any study by varying the mask percentage? How long are these other SSL tasks trained? Generally Contrastive and MAE tasks require a long training for better learning. Was this good practice followed?

**Questions:**

Questions are all detailed with the associated weaknesses raised above (in ‘weaknesses’ section). Here, in addition, some minor typos are mentioned.
- Line 156: Between ‘shifts’ and ‘In’, a stop is missing.
- Line 174: ‘… method, We …’ -> ‘method, we’
- Line 175: Appendix number is missing
- Line 178: ‘contains’ -> ‘containing’
- Line 200: ‘More baseline detailed are shown’ -> ‘More details about the baselines are provided’
- Line 204-205: ‘random initialize the kernel size k’ -> ‘randomly initialize the kernel of size k’

**Limitations:**

Limitations are not explicitly described. It may be useful to discuss the scope of the claims regarding the convolution operation in CVP handling structural bias in OOD.

---

> ### Author Rebuttal · Authors · 2023-08-09
>
> **Clarification for contrastive formulation**
>
> The loss indeed considers the negative pair, which is in the denominator. If $x_{i}$ and $x_{j}$ are positive pair, the $y_{i,j}$ is 1; then we maximize the similarity between the pos on the nominator and minimize the denominator, which is similarity between all negative pairs $x_{i}$ and $x_{k}$, where $j \neq k$. When $y_{i,j}$ is zero, the whole term is 0, and does not impact the loss.
>
> **Clarification for test-time adaptation framework**
>
> As the paper [A] defined, Test-time adaptation is running optimization at test time for target-domain data, which does not prohibit a model pre-trained with source data. Test-time adaptation methods, such as TENT, TTT, and MEMO, all train their model using the source data.
>
> [A] Gao, Yunhe, et al. "Visual prompt tuning for test-time domain adaptation." arXiv preprint arXiv:2210.04831 (2022).
>
> **What tells that the distribution shift is 'visually structured'?**
>
> Our visual world is known to be structured, and the changes are not random. For example, the shift introduced by blur is highly structured: you can tell there is blur in the image. This assumption works well according to our empirical results. We agree that our method may not handle arbitrary shifts that are not structured, but this is out of the scope of our paper and we leave this for future work.
>
> **Clarification Line 180+**
>
> - The conv. operation is applied to the image and only during the adaptation phase. We draw the flow of our proposed method in PDF Figure 1 in general responses.
> -  Yes, the MLP is used during the inference, yet the weight of it is fixed and not counted as learnable parameters at inference. When compared with other approaches, Fig. 6 shows only the number of learnable parameters at the inference time.
> - For Fig. 6, we redraw it and show it in the PDF Fig. 4. The x-axis is in unit K.
>
> **Line 184: When extending the proposed approach to CLIP**
> - We focus on the convolutional prompt only applicable to the vision encoder. Combining text prompts with CVP will be future work.
>
> **Clarification: Section 4.1 FT and PFT Baseline**
>
> Yes. Standard test-time adaptation initials or reset weights after a one-time adaptation, such as MEMO. Because the samples that arrived in batches can be uncorrelated to each other at inference, using the updated weights from prior batches will hurt the performance.
>
> **Line 206: What is a sharpness kernel?**
> - Sharpness is a standard $nxn$ matrix that enhances edges with a high pass filter. For example, [[0,-1, 0], [-1, 5, -1], [0, -1, 0]].
>
> **Compared to other TTA baselines**
>
> Our method is orthogonal to BN [53], TENT [60] and can be applied on top of them, improving those methods even more. Since our method updates only the input prompt without modifying the model weights, it works with frozen models and avoids the high cost of fine-tuning.
>
> **Details about low-rank prompts in ablation studies**
>
> 1. Pseudo algorithm for LVP. We will add this to the supplementary.
>  - SVD on input $X$ →  get initial $ U \Sigma V^T$
>  - Set rank number $r$ (ex: 3)
>  - For every adapt iteration $t$:
>     - Get inverse matrix $M_t = U_t * \Sigma_t * V^T _t$
>     - Apply low rank SVD on $M_t$ with rank number $r$ →  get low rank $U'_t$, $\Sigma'_t$, $V'^T _t$
>     - Get new inverse matrix $M’_t = U’_t * \Sigma’_t * V’^T _t$
>     - Get low rank X  → $X’ = X + M’_t$
>     - Calculate contrastive loss $L_s$ on $X’$ by inference the  SSL model
>     - Get gradient for $U'_t$,  $\Sigma'_t$, $V'^T _t$ and Update the three matrices via SGD
>
> 2. Are they directly applied to the input?
>     - Yes, the low-rank matrices are directly applied on input X
>
> 3. Clarification: How is it adapted to convolutional architectures?
>     - Since our low-rank prompt is applied on the input, not the intermediate layers of the model, it can be applied to convolutional neural networks following prior literature [B].
>     - [B] Yang, Yuzhe, et al. "Me-net: Towards effective adversarial robustness with matrix estimation." arXiv preprint arXiv:1905.11971 (2019).
>
> 4. Compare low-rank prompts with CVP
>    - We agree that low-rank prompts induce more learnable parameters due to the decomposition of matrices. However, compared with VP, CVP, and low-rank prompts require only 1\% and 9\% of the parameters, respectively, making both methods lightweight. Therefore, we want to highlight that both of these lightweight prompts have significant performance improvements.
>
> 5. We are sorry for the missing details. We already added a detailed explanation and full caption for Fig. 9 in our PDF Fig. 2.
>
> **Line 266-271 and Table 7, Questions about SSL tasks**
>
> 1. Sorry for the confusion. The 'Standard' means the backbone model without adding SSL. In Table 7, the standard model is the ResNet50 pre-trained on ImageNet. We compare the adaptation performance of three SSL tasks with this standard model.
> 2. Sorry for the typo. In Table 7, the reported results are the average robust accuracy, not the average error rate. As the subparagraph in the Ablation study mentioned, our CVP improves robust accuracy for every severity level on ImageNet-C when optimized with all three SSL tasks. We will add the details of SSL tasks in our later revision.
> 4. We have four degrees for our rotation prediction tasks, including 0, 90, 180, and 270.
> 5. We set up 75 \% of the images to be masked in MAE, following the default setting in [C] for setting up the mask percentage. We will add more studies on the varying mask percentages for MAE.
>     - [C] He, et al. "Masked autoencoders are scalable vision learners." CVPR'22.
> 6. As our MLP model is small, the SSL tasks take only 1~2 hours for training. We train SSL only once before making the adaptation. Once the SSL has been trained, it is fixed, and we don't need to tune it at test time. SSL module introduces little inference overhead in exchange for a large gain in robustness.
>
> **Thanks for mentioning the minor typo, we will fix it in our later revision**

---

> > ### Author Response · Authors · 2023-08-14
> >
> > Dear reviewer QKSZ,
> >
> > We appreciate your reviews and comments. We hope our responses address your concerns. In our rebuttal, we have added all the missing details and drew the flow for our proposed method (in general response PDF file). Please let us know if you have further question after reading our rebuttal. We hope to address all the potential issues during the discussion period.
> >
> > Thank you.

---

> > > ### Comment · Reviewer_QKSZ · 2023-08-17
> > > **Post rebuttal comments**
> > >
> > > Thanks for the detailed response. While some of the responses are helpful (clarification for contrastive formulation, what is a sharpness kernel etc.), some increased the confusion. For example, Figure 1 in general responses PDF. It provides the flow of the proposed method. It can be seen that during training source domain images (in-distribution) are used in the Training phase (left hand side of the figure). Then how can the proposed method be compared with testtime adaptation based methods like TENT. Table 1 in the TENT paper [60] tells that fully test time adaptation does not use source data (even if unlabeled). This is related to my earlier question in point 2 of the “weaknesses”.
> > >
> > > I also agree with reviewer zs9s and am equally concerned about the incremental nature of the work along with its performance in comparison to the approaches in table 4 (I also mentioned it in point 8 of the “weaknesses”). The response says that the proposed approach is orthogonal. However, “orthogonality” is not described enough. As I said previously, it seems these methods are less privileged than the proposed approach as they are not using source data. Now, going ahead with the argument that the proposed approach is different, the fact that combining it with the existing approaches results in only incremental performance improvement, does not convincingly say that the ‘differences’ are useful. The same table has the performance by ‘only’ the proposed approach (without combining it with any other existing approach) and this performance is pretty low compared to the existing approaches. So, whether combining the proposed approach with the existing ones is useful or combining the existing one with the proposed approach does a better job, is still debatable.

---

> > > > ### Author Response · Authors · 2023-08-17
> > > >
> > > > Thank you for replying.
> > > >
> > > > As mentioned above, we do not compare directly with those test-time adaptation-based methods like TENT, BN, or MEMO due to the different adaptation settings.  Table 4. which we show here, wants to convince the reader that our method can also complement those TTA methods. As our main goal considers a more challenging scenario than other TTA; that is, the model weight can not be adjusted during the test time. If under our setting, then most of the test-time adaption methods can not work because all of them need to change the model weight for every step. Therefore, we want to claim that ours are orthogonal to those methods and have benefits while the model weights are frozen. We already show our method has the SOTA performance in Table 1,2, and 3, compared to other visual prompts baselines, which only change the input weight and are more related to our settings.
> > > >
> > > > Besides, we want to clarify that those TTA methods also use the source data during training as they always need to have a pre-trained model that already learns the pre-knowledge from the source domain so that adaptation is allowable. For example, the TENT, BN, and MEMO, if they want to make a model adapt to the ImageNet-C, then they need a model that is well "pre-trained" on clean ImageNet, which is indeed the source domain data.

---

> > > > > ### Author Response · Authors · 2023-08-21
> > > > >
> > > > > Dear Reviewer QKSZ,
> > > > >
> > > > > We have tried our best to answer all the questions, and we sincerely hope you can fully understand. We look forward to receiving other feedback, and we deeply appreciate the time and effort you've dedicated to reviewing our work. Thank you

---

### Official Review · Reviewer_XDjm · 2023-06-26

**Soundness:** 3 good
**Presentation:** 4 excellent
**Contribution:** 3 good
**Rating:** 7
**Confidence:** 4

**Summary:**

This paper proposes a novel label-free approach CVP for test-time adaptation on out-of-distribution data. The main idea is to use a convolutional kernal as the visual prompt. It captures the structure of data distribution shift, and reduces the trainable parameters. Experiments show that CVP improves model robustness and complements existing weight-adaptation methods.

**Strengths:**

1. Prompt tuning has become an important technique to adapt large visual models. This paper provides a simple solution and may be useful in various applications.

2. The paper is well motivated. Using convolutional structure to inject the inductive bias is simple yet effective. It reduces the required parameters, and shows significant better performances than low-rank prompts.

3. The method does not require labeled data.

4. The method can be combined with other weight adaptation methods. It can also be generalized to multiple self-supervised objectives. The authors provide corresponding experiments with encouraging results.

5. The paper is well structured and nicely written. Empirical results are quite extensive. The attention visualization provides a clear insight.



**Weaknesses:**

1. The corruption types considered in this paper seem to be restricted to low-level transformations. This explains why the convolutional kernel is suitable. I wonder if CVP would still work well for high-level distribution shift, such as style changes?


Typo:
Reference missing in Ln 175.

**Questions:**

1. In Ln 206, "starting from a sharpness kernel is effective". Could the authors elaborate on what 'sharpness kernel' is and why it is effective?

2. The convolutional kernel applies on the whole image. Sometimes, the corruption may happen in some local regions, e.g., motion blur on foreground objects only. How do the authors comment on such situations?

**Limitations:**

The authors have not discussed limitations. But I feel that the method is simple, and experimental analyze are extensive enough.

It is helpful to discuss the effectiveness under other OOD types like open-class data, high-level style changes.

---

> ### Author Rebuttal · Authors · 2023-08-09
>
> **Thank you for endorsing and recognizing our work as realistic, interesting, and well-studied.**
>
> **CVP would still work well for high-level distribution shift, such as style changes?**
>
> - It's a very good point. We do have experiments on those style changes benchmark such as the ImageNet-Rendition, Sketch, and Adversarial. Our results show that CVP can improve robust accuracy on such a high-level distribution shift.
>
> **In L206., "starting from a sharpness kernel is effective". Could the authors elaborate on what 'sharpness kernel' is and why it is effective?**
>
> - When adapting, the kernel should be initialized with fix/random initialization. We use a sharpness kernel as an initial point for the fixed initialization setting, which is a n*n matrix. It starts from specific values, which can control the edge enhancement effect by extracting the frequency information of inputs with a high pass filter. When the kernel size is 3, we set up the sharpness kernel as [[0,-1, 0], [-1, 5, -1], [0, -1, 0]]
> - Similar to the convolutional kernel prompt, sharpness transforms an image using a convolutional 2D filter. It simultaneously enlarges the high-frequency part of images and then filters the low frequencies part.
>
> **Can convolutional kernel applies on the partial region**
> - This is an interesting point. We only apply it to the whole image. We will study this in future work.

---

> > ### Author Response · Authors · 2023-08-14
> >
> > Dear reviewer XDjm,
> >
> > We appreciate your reviews and comments. We hope our responses address your concerns. Please let us know if you have further question after reading our rebuttal. We hope to address all the potential issues during the discussion period.
> >
> > Thank you.

---

> > ### Comment · Reviewer_XDjm · 2023-08-16
> >
> > Dear Authors,
> >
> > Thanks for the response. I have carefully read that and other reviews.
> >
> > The response generally solves my previous questions on high-level shift and sharpness kernel. The proposed method works on ImageNet-R/S/A according to the experiments, though I agree with Reviewer uZKT that the motivation of using convolution should be made clearer, especially in the case of high-level shift. E.g., whether 'de-corruption operator' works for style-shift? Whether 'those shifts are often locally structured'?

---

> > > ### Author Response · Authors · 2023-08-21
> > >
> > > We thank reviewer XDjm for the responses.  We want to provide more clarification and discussion.
> > >
> > > Whether 'those shifts are often locally structured'? / Whether 'de-corruption operator' works for style shift data?
> > >
> > > As some literature [1, 2] suggests that part of the style shift is the change in the local information, such as textures, and part of it is the change in the global, such as sketches where object shapes are preserved yet texture cues are missing.
> > > Our prior experimental results show that CVP improves on several style-shift OOD benchmarks, such as ImageNet-Rendition, Sketch, and Adversarial. For the ImageNet-Rendition, the data contains several styles, such as art, cartoon and are more related to the local (texture) change. For the ImageNet-Sketch, the data contains all sketches that only preserve object shapes and are more related to global change.
> > > Therefore, the de-corruption operator like CVP can work on both local / global change of style shifts.
> > >
> > > [1] Geirhos, Robert, et al. "ImageNet-trained CNNs are biased towards texture; increasing shape bias improves accuracy and robustness." arXiv preprint arXiv:1811.12231 (2018).
> > > [2] Wang, Haohan, et al. "Learning robust global representations by penalizing local predictive power." Advances in Neural Information Processing Systems 32 (2019).

---

### Official Review · Reviewer_HEpT · 2023-07-08

**Soundness:** 2 fair
**Presentation:** 3 good
**Contribution:** 4 excellent
**Rating:** 6
**Confidence:** 4

**Summary:**

.The paper proposes a new method for test-time adaptation (TTA) to structured distribution shifts. The proposed method learns convolutional visual prompts to prevent the model from overfitting to SSL objectives due to high dimensional prompts. The results show that the proposed approach consistently results in performance improvements for TTA to OOD variants of CIFAR-10 and ImageNet.

**Strengths:**

- The idea of using convolutional prompts to avoid SSL overfitting is novel and well-motivated (Table 5 shows how SSL results in overfitting and Figure 3 shows the need for some form of inductive bias).
- The results indicate consistent results when CVP are added to baselines and other prior TTA works.
- The method tunes very few parameters. The parameter efficiency allows it to be adopted in edge computing devices working with low memory and compute.

**Weaknesses:**

1. The paper compares their method with only two types of visual prompts (patch and padding based [40]). The paper does report results with CLIP ViT-B (Table 3), but does not compare to other prompts used for adapting ViTs [26] and text-only prompts for CLIP-pretrained ViTs. [A]
2. The paper mentions on L63: “The only one that does not update the model (for Test time adaptation) is proposed by [40]”. However the following uncited works do not update the model as well [A, B].
3. The paper does not show results with unimodal ViT architectures (eg. pretrained models from TIMM). Similar gains over the baselines (and [26]) in the context of ViTs can boost the applicability of the approach.
4. The paper sets different hyperparameters for different datasets (range of $\lambda$, kernel sizes), without providing guidelines on how these are to be determined for a new dataset (or corruption type).
5. Missing details on evaluation and implementation (covered under Questions). Minor writing: $\lambda$ is introduced much later in text, while being introduced in 3.3.

[A] Test-Time Prompt Tuning for Zero-Shot Generalization in Vision-Language Models. Shu et al.

[B] Test-Time Training with Masked Autoencoders. Yossi et al.

[C] Dual Modality Prompt Tuning for Vision-Language Pre-Trained Model. Xing et al.

**Questions:**

I am happy to update my rating if the concerns around comparisons to prior work and hyperparameter selection are adequately addressed. But I request the authors to provide the missing details.

1. In Table 8, multiple choices are provided for kernel size and update iters, how are the chosen hyperparameters validated?
2. What are the hyperparameters used in Table 4/5/7, where the kernel size is not specified?
3. How is CVP applied to CLIP-ViTs? Is the image first convolved and then tokenized?
4. Which CLIP-ViT/32 model and can was used for the results in Table 3? [A] does report numbers on CLIP-ViT-base/16.
5. What architecture is used for the results in Table 7?
6. For the MAE baseline in Table 7: is the contrastive objective of CVP directly replaced by MAE objective? But this introduces additional decoder parameters as well right? It would be good to see a comparison of the number of parameters that are optimized in different SSL methods.
7. L9: “1% when compared to standard visual prompts”. How does this compare to the shallow prompts used in [26].
8. Tables 9 and 12 show that CVP does not outperform baselines for fog/frost/snow corruptions. Also, Figure 3 shows CVP does not outperform low-rank prompts for snow corruption. Any intuition on why CVP does not perform well when weather conditions are changed?
9. In Table 15, are the reported losses on samples that were used to tune prompts? Reporting the loss on unseen samples may give a better idea of the extent of overfitting.
10. Since [A] is a prior TTA work for CLIP, would there be any benefits of dual text-visual prompt tuning (like [C]) with: CVP (your method) + [A] (NeurIPS’22)?


[A] Test-Time Prompt Tuning for Zero-Shot Generalization in Vision-Language Models. Shu et al. (NeurIPS'22)

[B] Test-Time Training with Masked Autoencoders. Yossi et al. (NeurIPS'22)

[C] Dual Modality Prompt Tuning for Vision-Language Pre-Trained Model. Xing et al.

**Limitations:**

The paper may need to add a limitation (and future work) section if the above concerns (applicability to uni-modal ViTs and hyperparameter tuning) can not be addressed in the current submission.

---

> ### Author Rebuttal · Authors · 2023-08-09
>
> **We appreciate the reviewer’s comments and suggestions. Thanks to the reviewer for recognizing the novelty of our work. We have answered and addressed the questions.**
>
> **Comparison with other prompts**
>
> - Thank you for your suggestion of comparing with other prompts. We have followed your suggestion and included an additional comparison with the shallow prompt [26] in the Table below.
> - We mention that [A] uses text prompts tailored to vision language model adaptation. Our method is more general and can be applied to any visual perception model.
> - We run the experiment on CIFAR10-C and show robust accuracy. The shallow prompt doesn't improve under our setting because it requires more learnable parameters in the prompt and leads to overfitting for adaptation.
> - |                | s1    | s2    | s3    | s4    | s5    |
> |----------------|-------|-------|-------|-------|-------|
> | CLIP-ViT-b/32  | 58.58 | 48.45 | 40.12 | 33.38 | 27.51 |
> | Shallow prompt [26] | 56.87 | 47.22 | 38.72 | 30.49 | 24.10 |
> | CVP            | 59.11 | 49.09 | 40.76 | 33.51 | 27.80 |
>
> **Other test-time adaption method**
>
> Thanks to the reviewer for pointing out those works [A] [B]. We will definitely cite them in our revision.
>
> [A] Test-Time Prompt Tuning for Zero-Shot Generalization in Vision-Language Models. Shu et al.
> [B] Test-Time Training with Masked Autoencoders. Yossi et al.
>
> **ViT Results**
> - Thanks to the reviewer for suggesting this. We ran the experiment for CVP on the ViT-Base model with results showing a similar gain in performance.
> - The following table shows our results of the ViT-Base model for 15 types of corruption under severity level 1 on CIFAR-10-C. The test accuracy of ViT-Base on CIFAR10 is 96.42\%. Compared to the accuracy (\%) before adaptation (standard) and using VP to do the adaptation, CVP achieves up to 2% better performance. More detail of ViT results can be found in the PDF Table 1. in general responses.
>
> - |                   |          | ViT-Base/32 |       |
>   |-------------------|----------|:--------:|-------|
>   |                   | standard |    VP    |  CVP  |
>   | Averaged Acc. | 66.45 | 66.48 | **67.77** |
>
>
> **The choice of hyper-parameter setting on kernel size, $\lambda$ range, and update iterations**
> - For the $\lambda$ parameter, it controls the magnitude of convolved output when combined with the residual input. We set the $\lambda$ range to be [0.5, 3] and run test-time optimization to automatically find the optimal solution, which does not require a validation set.
>
> - We use 3x3 for cifar-10 and 5x5 for ImageNet. In general, small kernel size is used to avoid overfitting. We increase the kernel size for large images, such as ImageNet.
>
> - For the update iterations, as the Table 17. show a larger number of iterations has better performance. However, the training cost becomes higher.
>
> **Missing details on evaluation and implementation**
> - Thanks for mentioning it. We will add all of these details in our later revision.
> 1. How are the chosen hyperparameters validated in Table 8?
>     - See the above section, "The choice of hyper-parameter setting."
> 2. What hyperparameters are used in Table 4/5/7, where the kernel size is not specified
>     - For Table 4, we set the kernel size for CIFAR-10-C as 3x3 and ImageNet-C, R, S, and A as 5x5. The adaptation iterations are all set as 5.
>     - For Table 5, same as the setting above, the CIFAR-10-C uses the 3x3 and others use 5x5. The adaptation iterations are all set as 5.
>     - For Table 7, we evaluate the performance for three SSL tasks on ImageNet-C. The standard model is ResNet50. For the CVP kernel, we use the same setting as above, the 5x5, which is the optimal choice.
> 3. How is CVP applied to CLIP-ViTs? Is the image first convolved and then tokenized?
>    - Sorry for missing details. Yes, our CVP is applied before tokenized. The CVP is applied to CLIP-ViTs by adding a convolutional kernel on the input sample and then getting an adapted sample as a new input iteratively. In our later revision, we will add more detail on how CVP applies to CLIP.
> 4. Which CLIP-ViT/32 model was used for the results in Table 3?
>     - Thanks for the careful check. The CLIP model we use for evaluation is the CLIP-ViT-base/32. In our later revision, We will clarify our model CLIP-ViT/32 as CLIP-ViT-base/32.
> 5. What architecture is used for the results in Table 7?
>     - The architecture we use for Table 7 is the ResNet50 pre-trained with original ImageNet-C.
> 6. Clarification for MAE baseline
>     - Yes,  the contrastive objective of CVP is directly replaced by the MAE objective. We replace the contrastive loss with reconstruction loss.
> Since the decoder for MAE is fixed, we do not optimize additional parameters at test-time adaptation.
> 7. L9: 1\% when compared to standard visual prompts”. How does this compare to the shallow prompts used in?
>    - Thank you for your question. We compared to shallow prompts, which is also less than 1\%.
> 8. Why does CVP not outperform baseline on weather conditions
>    - While our CVP doesn't outperform the VP baseline on weather corruption, it outperforms the standard baseline results.
>    - In Figure 2., we show deep insight into the contrastive loss distribution and empirically find that the weather corruption (for example, snow) leads to a huge shift from the source domain even in the low severity. Therefore, we conjecture that the weather corruption types over-damaged the intrinsic structure and need more learnable parameters to restore the structure.
> 9. In Table 15, are the reported losses on samples used to tune prompts?
>     - Yes, we report the losses for samples already tuning with the prompt.
>     -  We run the loss analysis on unseen test samples and show CVP can avoid overfitting. (shown in PDF Fig. 3 in general responses).
> 10. Dual text-visual prompt tuning
>     - Thank you for these suggestions. We feel that combining text prompts and CVP is an interesting idea. We will add it in future work.

---

> > ### Author Response · Authors · 2023-08-14
> >
> > Dear reviewer HEpT,
> >
> > We appreciate your reviews and comments. We hope our responses address your concerns. In our rebuttal, we have added all the missing details and added the experiment, including ViT results, shallow prompts baselines on CLIP, and the loss analysis for unseen samples (in our PDF file). Please let us know if you have further question after reading our rebuttal. We hope to address all the potential issues during the discussion period.
> >
> > Thank you.

---

> > > ### Comment · Reviewer_HEpT · 2023-08-18
> > > **Thanks for the rebuttal**
> > >
> > > I thank the authors for the additional experiments and clarifications provided. I appreciate the comparison to shallow prompts, clarification of low performance on weather perturbations, and the loss analysis on unseen samples. I am generally satisfied with the responses, which suggest that CVP is a better-prompting strategy compared to prior prompting strategies when dealing with structured (eg. blur) and style perturbations (eg. sketch, renditions). But after carefully going through all reviews and rebuttals, I have a few remaining concerns and encourage authors to provide clarifications on these:
> > >
> > > > The shallow prompt doesn't improve under our setting because it requires more learnable parameters in the prompt and leads to overfitting for adaptation.
> > >
> > > Are the improvements with CVP (here, and in general) only due to much less learnable parameters? Is there a direct comparison with prompts having exactly the same number of parameters as CVP (eg. partially learnable prompts) that shows the importance of using convolutional structures? I thank reviewer QKSZ for pointing out that the low-rank prompts, although lightweight, actually have a higher number of parameters than CVP.
> > >
> > > > We feel that combining text prompts and CVP is an interesting idea. We will add it in future work.
> > >
> > > It would have been nice to show CVP leads to improvements when combined with text prompts as well, but leaving it to future work sounds good to me. In the existing comparisons that show improvements over prior work - do the authors use the SSL objective during source training for the MEMO, BN and TENT methods? Do the MEMO and MEMO + CVP rows start with the same source model? If not, is it possible that the SSL pretraining on source gives an unfair advantage to MEMO+CVP compared to MEMO?

---

> > > > ### Author Response · Authors · 2023-08-18
> > > >
> > > > Thank the reviewer HEpT for carefully reading our rebuttal and giving us valuable feedback. We would like to provide more clarification here.
> > > >
> > > > 1. Regarding the improvements of CVP, it is not just less learnable parameters but also the convolutional operator that helps to avoid overfitting. Without the convolutional kernel, the structure can not be captured well during the adaption. In Tables 1 and 2, we show the results on SVP (patch), which is the baseline without convolution structure and only adds the $\ell_p$-norm perturbations as the same shape of input. We agree that the results compared with the same number of parameters as CVP are necessary, and we thank both reviewers HEpT and QKSZ for pointing out this. Therefore, we add another experiment to compare the CVP with SVP (patch) under different learnable numbers of parameters. We show robust accuracy on the CIFAR-10-C, including average of 15 types of corruption with five severities. The model backbone for Standard results is WideResNet18.
> > > >
> > > > - |                  | S1     | S2     | S3     | S4     | S5     | Avg. Acc. |
> > > > |------------------|--------|--------|--------|--------|--------|-----------|
> > > > | Standard     | 59.68% | 47.88% | 40.31% | 32.75% | 28.20% | 41.76%    |
> > > > | SVP (patch 3*3)   | 59.71% | 47.90% | 40.34% | 32.76% | 28.23% | 41.79%    |
> > > > | **CVP**              | **68.07%** | **54.73%** | **45.96%** | **37.79%** | **31.61%** | **47.63%**    |
> > > >
> > > > - Here, we set up the patch size as 3x3. In the third row of the table, the SVP (patch 3x3) has the same learnable number of parameters when CVP uses the 3x3 convolutional kernel. When with the same number of parameters, CVP has improved the performance by 5.9 points which indicates the importance of using convolutional kernels.
> > > >
> > > > 2. Regarding the existing comparisons that show improvements over prior work, do the authors use the SSL objective during source training for the MEMO, BN, and TENT methods?
> > > > - For MEMO, BN, and TENT methods, we do not use the SSL objective during source training. The source model is frozen all the time. Besides, the SSL model that we train for the purpose of combining with MEMO, BN, and TENT will not affect the source model.
> > > >
> > > > 3. Do the MEMO and MEMO + CVP rows start with the same source model? If not, is it possible that the SSL pretraining on source gives an unfair advantage to MEMO+CVP compared to MEMO?
> > > > - Yes, MEMO and MEMO + CVP indeed start from the same source model. As our flow figure show in the PDF, we don't need to retrain the source model and only require to train an auxiliary SSL model based on the frozen source model. The SSL model on the source will not affect the source model or change any pretrained weight in the source model. During the test time, for every adaptation step, the MEMO + CVP first optimizes the convolutional kernel with SSL objective to generate a new sample and then use the new sample for adapting the model weight. Therefore, the comparison is fair.
> > > >
> > > > We have tried our best to answer all the questions, and sincerely hope the reviewer can fully understand. We look forward to receiving other feedback and we deeply appreciate the time and effort you've dedicated to reviewing our work.

---

> > > > > ### Comment · Reviewer_HEpT · 2023-08-19
> > > > > **Thanks for the clarifications**
> > > > >
> > > > > I thank the authors for the clarifications. I am happy to increase my rating because the authors have addressed all my concerns. I strongly encourage the authors to update the paper with the new results and figures.

---

> > > > > > ### Author Response · Authors · 2023-08-21
> > > > > >
> > > > > > Dear Reviewer HEpT,
> > > > > >
> > > > > > Thanks for your valuable comments on our paper and for deciding to raise the rating. We will update the paper with all implementation details, new results, and figures in the upcoming version.
> > > > > >
> > > > > > Thanks a lot for your time again!

---

### Official Review · Reviewer_zs9s · 2023-07-28

**Soundness:** 2 fair
**Presentation:** 2 fair
**Contribution:** 1 poor
**Rating:** 4
**Confidence:** 4

**Summary:**

The paper proposes a variant of Visual Prompt Tuning (VPT) where the prompt is applied to a penultimate layer of the encoder network and is a result of a convolution with a 3x3 or 5x5 kernel (essentially it is an added residual block or residual adapter if you will). The added adapter is trained at test time by applying a contrastive loss on a test batch (so, in fact, it seems to be a variant of transductive learning, requiring a batch of test images - to form a negative set for the contrastive loss, so not truly online and does not seem to be able to operate on a single test sample). The authors test their approach on a collection of OOD benchmarks, some synthetic, and some real - these ones are in fact UDA / UDG benchmarks, however, UDG baselines are not compared. Small improvements are observed over naive baselines, though prior methods like [53] and [60] seem to deliver much better performance (Table 4) and the authors only show the benefit of their approach via ensembling / combining with the other methods.

**Strengths:**

- test time adaptation, VPT, and handling domain shifts are important topics
- some improvement is observed over simple baselines

**Weaknesses:**

- novelty: adding a convolutional residual adapter + contrastive loss for transductive learning (test time adaptation on a batch) is hardly novel
- comparisons to prior works:
* table 4 shows [53] and [60] attain significantly better results than the proposed approach
* relevant baselines / benchmarks seem to be missing (e.g. https://arxiv.org/pdf/2210.04831.pdf reports better results on ImageNet-C + DomainNet benchmark could be added)
* I would expect comparison to / on top of UDG baselines that solve the same problem
* combining methods as in table 4 is fine, but other methods could also be combined, would it yield even higher results?
- Supervised variant of the approach explored in the ablation needs to be compared to few-shot and in particular transductive few-shot methods, expecting much higher results there.
- what about the online test-time tuning (from one sample) seems not possible due to the use of the contrastive loss?


**Questions:**

- PromptTuning is a weaker variant of the broader family of PEFT methods, usually applied on the input level (penultimate layer in this case) it does not have enough representation power to model more complex deviations, as opposed to more popular LoRA or prefix tuning methods. Would the authors' approach be applicable to the more powerful PEFT methods?

**Limitations:**

limitations not discussed

---

> ### Author Rebuttal · Authors · 2023-08-09
>
> **Novelty of our work**
> - Our paper provides deep insights on what is a good visual prompt design for test-time adaptation, which is an important problem. The key novelty of our work is to propose this simple and effective convolutional visual prompt to address the overfitting challenge at test time.
> Prior work like convolutional residual adapter adapts the model architecture with lots of parameters, which is designed for training time optimization but can lead to overfitting on test-time adaptation. Our method only adds a small convolutional prompt on the inputs during the inference time, which is both lightweight and avoids test-time overfitting.
> - Moreover, the other Reviewers XDjm and HEpT find our method to be novel and well-motivated. Reviewer uZKT also mentions this research direction would interest the visual prompt community.
>
> **Performance compared with other methods (BN [53] and TENT [60])**
>
> - Our method is orthogonal to BN [53], TENT [60] and can be applied on top of them, improving those methods even more. Since our method updates only the input prompt without modifying the model weights, it works with frozen models and avoids the high cost of fine-tuning (especially on edge devices like smartphones). Our work provides a new direction for the robustness field by visual prompting at test time, which we think is worth publishing.
>
> **Compare with relevant baselines such as [A]**
>
> - We thank the reviewer for suggesting this relevant paper, which we will cite.
>
> - We highlight the difference between our work and this paper as follows. 1.) Our CVP creates during the adaptation phase, and we don't need any source domain sample to tune the prompt first. 2.)  In contrast to [1] tuning a collection of visual prompts and the classification head, we adapt every batch sample using CVP, which is just a single convolutional kernel. Our method is simple, lightweight, and easy to use.
>
> - [A] Gao, Yunhe, et al. "Visual prompt tuning for test-time domain adaptation." arXiv preprint arXiv:2210.04831 (2022).
>
> **Comparison to / on top of UDG baselines that solve the same problem**
>
> - Standard UDG methods require training on multiple source domains and testing generalization on a different domain, which is not the ImageNet benchmark setup we evaluate on.
> - Our UDG scenario has only a single training domain, like Imagenet, and tests the model on multiple OOD domains. We thus only compare UDG methods that conduct single-domain training, such as BN, TENT, and MEMO. Experimenting with UDG with multiple training domains is out of the scope of our paper.
> - We would like to include more baselines if the reviewer can point out other single training domains UDG work that we are missing.
>
> **Combining other methods**
> - Thank you for your question. We agree that since our method applied a prompt for the input, it can be easily combined with other methods that change the model weights. We believe our method can further improve the robustness of future work that adapts model weights.
>
> **Compare the supervised variant of the approach with transductive few-shot**
> - We want to clarify that results from the supervised variant are only for training the best case and thresholding the upper bounds of our method.
> - Our goal is to focus on test-time adaptation without the annotations, where we indeed compare with the transductive few-shot method (For example: MEMO) under the self-supervised setting (see Table 4.). The supervised variant of the approach represents the baseline which is not our main contribution.
>
> **The online test-time tuning with one sample**
>
> - Our convolutional visual prompt is general and can be applied to other SSL tasks that use one sample. In Table 7, we show our method also works with rotation prediction and MAE, which only needs one sample and achieves up to 3.14\% improvement in robustness.
>
>
> **Can CVP complement other parameter-efficient tuning (PEFT) methods?**
>
> - Yes, they are complementary. The PEFT methods often update the model weights, while our CVP updates the input. Thus our method is orthogonal to PEFT and can be directly applied to PEFT to improve the model adaptation performance further. We will add this in our future work.

---

> > ### Author Response · Authors · 2023-08-14
> >
> > Dear reviewer zs9s,
> >
> > We appreciate your reviews and comments. We hope our responses address your concerns. Please let us know if you have further question after reading our rebuttal. We hope to address all the potential issues during the discussion period.
> >
> > Thank you.

---

> > ### Comment · Reviewer_zs9s · 2023-08-16
> > **post rebuttal**
> >
> > thank you for your response. Reading the rebuttal, I still have concerns:
> > 1. regarding comparison with Gao et al. 2022 - stating the difference does not change the fact it is a higher performance baseline. If prompt pre-training is seen as an advantage, I urge the authors to show their method could be used to improve on top of Gao et al. result by using prompt pre-training etc. Or adding CVP to the Gao et al. method and showing further improvement.
> > 2. Regarding CVP only improving over other methods when combined with them - my concern was - other methods can also be combined, and also standard VPT can be added to other methods - all those are additional baselines the authors need to show gains over to establish the importance of CVP design. Also, in most cases, the combination gains of CVP with other methods seem rather small (Table 4) and I wonder if it is significant enough.
> > 3. for UDG baselines - many UDG methods could be employed with single-domain pre-training. In fact, some are evaluated in source-target pairs as a standard benchmark - I think a comparison to some of those could be shown, but perhaps concern 2 above is more important
> > 4. for single sample vs transductive - if a single sample method can perform better than (transductive) contrastive approach - it should have been featured in all other experiments besides Table 7, if not - transductive baselines should have been used - from my experience, transductive inference (offline prediction for several test samples at once allowing them to learn from each other without any label knowledge) always adds a significant (could be even 5% or more) improvement to the results, and I think it is covering the improvement range observed in the current paper.
> > In light of the above, I prefer to keep my original rating.

---

> > > ### Author Response · Authors · 2023-08-17
> > >
> > > Thanks for your replying
> > >
> > > 1. Regarding comparison with Gao et al. 2022
> > >  - We thank the reviewer for pointing out this amazing work. However, we also realize this work is still unpublished and under review. Therefore, we will consider comparing with them in future work.
> > >
> > > 2. Regarding CVP only improving over other methods when combined with them
> > > - We are currently running the experiment of combining standard VPT with other TTA methods. We will report the numbers as soon as possible.

---

> > > > ### Author Response · Authors · 2023-08-19
> > > >
> > > > Dear Reviewer zs9s, thank you for taking the time to review our paper.
> > > >
> > > > We want to provide more clarification and experimental results for question 2.
> > > > - Regarding CVP only improving over other methods when combined with them, we ran another experiment to combine the standard VPT with other TTA methods, including BN, TENT, and MEMO. Our evaluation is on the CIFAR-10-C average on 15 corruption types and five severities. The results show that as below, in general, CVP + others have a lower error rate when on top of those three TTA baselines compare to VP + others.
> > > >
> > > > |             | Avg. Error Rate |
> > > > |-------------|:-----------------|
> > > > | source-only |           58.84 |
> > > > |  BN          |          38.51% |
> > > > | BN + VP     |          38.50% |
> > > > |**BN + CVP**     |          **38.39%** |
> > > > | TENT        |          38.52% |
> > > > | TENT + VP   |          37.65% |
> > > > | **TENT + CVP**  |          **36.69%** |
> > > > | MEMO        |          56.13% |
> > > > | MEMO + VP   |          55.43% |
> > > > | **MEMO + CVP**  |          **54.85%** |

---

> > > > > ### Author Response · Authors · 2023-08-21
> > > > >
> > > > > Dear Reviewer zs9s,
> > > > >
> > > > > We have tried our best to answer all the questions, and we sincerely hope you can fully understand. We look forward to receiving other feedback, and we deeply appreciate the time and effort you've dedicated to reviewing our work. Thank you

---

### Official Review · Reviewer_uZKT · 2023-07-31

**Soundness:** 3 good
**Presentation:** 3 good
**Contribution:** 3 good
**Rating:** 6
**Confidence:** 4

**Summary:**

- The paper introduces Convolutional Visual Prompts (CVP), a novel methodology designed to increase the model robustness when faced with Out-of-Distribution (OOD) data during test time.
- The primary innovation of CVP is the use of convolutional structures as inductive biases for adapting to visual OOD instances during test time.
- The experimental results show that these convolutional structures are efficient and effective at dealing with OOD corruptions.

**Strengths:**

- *Originality*: The paper proposes a novel method, the use of convolutional structures as inductive biases for visual prompt tuning to OOD data. This work would interests the visual prompt community.
- *Quality*: Extensive experiments have been conducted across a variety of OOD recognition tasks, providing empirical support for the effectiveness of CVP.
- *Clarity*: The paper is well-written and comprehensible, offering clear explanations of the problem, the proposed solution, and the obtained results.

**Weaknesses:**

The major concern of this work is its Motivation behind the choice of convolutional structures: The paper does not provide a clear motivation or justification for choosing convolutional structures as the basis for the Convolutional Visual Prompts (CVP). While the paper mentions the effectiveness of convolutional operations for handling structured data with local motifs, a more explicit link between this principle and the task of OOD adaptation would be beneficial. For instance, the authors could hypothesize that the learned convolutional kernel might act as the de-corruption operator for specific corruptions, offering a possible reason for their choice (the point was raised by the visual evidence like Figure 11, which seems indicate the reason behind why CVP works).

**Questions:**

- What motivated the choice of convolutional structures over other possible structures for the CVP?
- Could you discuss the wider usage of CVP when applied to settings other than test-time adaptation?
- Could you also discuss the extendability of CVP model, like the possibility/effectiveness of adding more kernels? I am not saying to develop a new variant during the rebuttal, but to indicate how the followers can use the method to the best extend would enhance the significance of the work.

**Limitations:**

The paper seems not discuss the limitations, but when the authors try to answer the weakness and questions above, they might come up with more discussion on the limitation of the work.

---

> ### Author Rebuttal · Authors · 2023-08-09
>
> **Motivation of using convolutional visual prompt for test-time adaptation**
>
> Thank you for your question. We agree with the reviewer that our CVP can be viewed as a de-corruption operator for natural images. Our paper studies how to be robust under major natural distribution shifts, such as style, smoothness, or lightening changes. Those shifts are often locally structured, which motivates us to use convolution that handles locally structured data. Empirically, our convolution directly undo the corruption, as shown in Figure 11.
> Moreover, our empirical results show that our proposed convolutional visual prompts outperform standard visual prompts and low-rank visual prompts. We will make the input de-corruption operation point more clear in our revision. It will be interesting to explore how well CVP adapts to more global changes, such as shape change, which we leave for future work.
>
> **Discuss the wider usage of CVP when applied to settings other than test-time adaptation**
>
> We thank the reviewer for pointing out this. We will add it in the discussion part for the later revision. We list a few additional usages of CVP below:
>
> 1. Beyond unsupervised test-time adaptation, we can also supervisedly optimize our convolutional visual prompt when a few labeled data are available on the target domain. Since our convolutional visual prompt has fewer parameters, applying it to this setting can reduce the training cost.
>
> 2. Continual learning can benefit from our lightweight CVP. Since the environment often changes incrementally, applying the structured CVP adaptation can potentially help continual adaptation without forgetting.
>
> 3. The model deployed on edge devices can also benefit from CVP. As it has limited memory and requires handling multiple types of corruption, the efficiency of the parameter of CVP allows us to maintain individual prompts for each scenario without using too much memory.
>
> **Discuss the extendability of CVP model**
>
> Thank you for your question. Yes, our method can be extended by stacking deeper convolutional prompts. In addition, future work can study adding CVP to the latent representations and study if the hyper network can be used to generate the parameters for the CVP for meta-learning. In terms of application, CVP can be used to adapt not only classification models but also generative models such as diffusion.

---

> > ### Author Response · Authors · 2023-08-14
> >
> > Dear reviewer uZKT,
> >
> > We appreciate your reviews and comments. We hope our responses address your concerns. Please let us know if you have further question after reading our rebuttal. We hope to address all the potential issues during the discussion period.
> >
> > Thank you.

---

> > > ### Comment · Reviewer_uZKT · 2023-08-21
> > > **Post-Rebuttal Comment**
> > >
> > > Overall, the authors have addressed the primary concerns and clarified the underlying motivations and potential extensions of their work. The broader applications of CVP, as highlighted in the rebuttal, make this work even more compelling. So I keep my original score.

---

### Author Rebuttal · Authors · 2023-08-10

- We thank the reviewers for the constructive feedback and insightful questions. We are delighted that most reviewers like the well-motivated novel method of CVP and think it would interest the visual prompt community, the extensive experiments conducted across various OOD recognition at large scale, and the clear writing. We address two common questions here.


 **Q1. Motivation of convolutional visual prompt (CVP)**
- Our paper systematically studies how to be robust under major natural distribution shifts, such as style, blurring, or lighting changes. Those shifts are often locally structured, which motivates us to use the convolutional kernel to handle locally structured data. Besides, the traditional visual prompts (VP) often induce too many learnable parameters, which causes the overfitting problem. Therefore, we propose a new structured prompt, the CVP. Our extensive experimental results show that CVP is better than traditional VP, which is more efficient and lightweight.

**Q2. Performance comparison with other test-time adaptation baselines**
- The traditional TTA methods usually assume the model weights can be adjusted during the test time. However, we want to claim that our method is orthogonal to those traditional TTA methods, such as TENT, MEMO, and BN, where we consider a more challenging scenario and assume the model weights are frozen during inference time. We thus demonstrate how our method can be combined with traditional TTA. Combining them shows that our method complements theirs and has improved on several OOD benchmarks.

We address the questions in individual responses and include additional experiments and figures to support our responses. We will add all the details to our later revision.  The figures and tables can be found in the PDF file.

Thank you again for your efficient handling of our submission.

---

### Decision · Program_Chairs · 2023-09-21

**Decision:**

Accept (poster)

**Comment:**

The final rating for this paper were mixed with one accept, two weak accepts, one borderline reject and one reject. The reviewers had a thorough rebuttal/discussion with the authors that resulted in many questions being resolved and one reviewer (HEpT) raising their score to weak accept.  The key strengths of the work are the focus on an important problem with a paper that is clearly written an extensive experiments. There was disagreement among the reviewers about whether the proposed method which introduces a convolutional visual prompt for test time training has sufficient novelty to warrant acceptance. After considering all reviewer and author comments / discussion the concerns about limited novelty are overshadowed by the fact that this work introduces a simple and potentially useful approach for test time adaptation with extensive experiments in a well written form factor.